



# Raindrop Fall Velocities from an Optical Array Probe and 2D-Video Disdrometer

Viswanathan Bringi[1], Merhala Thurai[1] and Darrel Baumgardner[2]

[1] *Department of Electrical and Computer Engineering, Colorado State University, Fort Collins, Colorado, USA*

[2] *Droplet Measurements Technologies, Longmont, Colorado, USA*

Correspondence to: V.N. Bringi

Email: bringi@colostate.edu



## Abstract

We report on fall speed measurements of rain drops in light-to-heavy rain events from two climatically different regimes (Greeley, Colorado, and Huntsville, Alabama) using the high resolution (50 microns) Meteorological Particle Spectrometer (MPS) and a 3$^{rd}$ generation (170 microns resolution) 2D-video disdrometer (2DVD). To mitigate wind-effects, especially for the small drops, both instruments were installed within a 2/3-scale Double Fence Intercomparison Reference (DFIR) enclosure. Two cases involved light-to-moderate wind speeds/gusts while the third case was a tornadic supercell that passed over the site with high wind speeds/gusts. As a proxy for turbulent intensity, maximum wind speeds from 10-m height at the instrumented site recorded every 3 s were differenced with the 5-min average wind speeds and then squared. The fall speed versus size from 0.1-2 mm were derived from the MPS data and the 2DVD was used for sizes >0.7 mm. Consistency of fall speeds from the two instruments in the overlap region (0.7-2 mm) gave confidence in the data quality and processing methodologies. Our results indicate that under light-to-moderate wind gusts, the mean fall speeds agree well with fits to the terminal velocity measured in the laboratory by Gunn and Kinzer from 100 microns up to precipitation sizes. In the supercell case the very strong gusts and inferred high turbulence intensity caused a significant broadening of the fall speed distributions with the mean fall speeds about 25-30% less than the terminal velocity of Gunn-Kinzer, i.e. sub-terminal fall speeds.



# 1 Introduction

Knowledge of the terminal fall speed of raindrops as function of size is important in modelling collisional break-up and coalescence processes (e.g., *List et al.,* 1987), in the radar-based estimation of rain rate, in retrieval of drop size distribution using Doppler spectra at vertical incidence (e.g., *Sekhon and Srivastava,* 1971) and in soil erosion studies (e.g., *Rosewell* 1986). The terminal velocity measurements of *Gunn and Kinzer,* 1949) under calm laboratory conditions, and fits to their data (e.g., *Atlas et al.,* 1973; *Foote and du Toit,* 1969; *Beard and Pruppacher,* 1969) are still considered the standard against which measurements using more modern optical instruments in natural rain are compared (*Löffler-Mang and Joss,* 2000; *Barthazy et al.,* 2004; *Schönhuber et al.*, 2008; *Testik and Rahman,* 2016). More recently, the broadening and skewness of the fall speed distributions of a given size (3 mm) in one intense rain event was attributed to mixed-mode amplitude oscillations (*Thurai et al.*, 2013). Super- and sub-terminal fall speeds in intense rain shafts have been detected and attributed, respectively, to drop breakup fragments (sizes < 0.5 mm), and high wind/gusts (sizes 1-2 mm) (*Montero-Martinez et al.*, 2009; *Larsen et al.,* 2014; *Montero-Martinez and Garcia-Garcia*, 2016).

The fall speeds and concentration of small drops (< 1 mm) in natural rain are difficult to measure given the poor resolution (>170 microns) of most optical disdrometers and/or sensitivity issues. While cloud imaging probes (with high resolution 25-50 microns) on aircraft have been used for many years they generally cannot measure the fall speeds. A relatively new instrument, the Meteorological Particle Spectrometer (MPS) is a droplet imaging probe that was built by Droplet Measurements Technologies (DMT, Inc.) under contract from the US Weather Service specifically designed for drizzle as small as 50 μm and rain drops up to 3 mm. This instrument in conjunction with a lower resolution 2D-Video Disdrometer (*Schoenhuber et al.,* 2008) is used in this paper to measure fall speed distributions in natural rain.

This paper briefly describes the instruments used, presents fall speed measurements from two sites under relatively low wind conditions, and one case from an unusual tornadic supercell with high winds and gusts and ends with a brief discussion and summary of the results.

# 2 Instrumentation and Measurements

The principal instruments used in this study are the MPS and 3$^{rd}$ generation 2D-video disdrometer (2DVD), both located within a 2/3-scale Double Fence Intercomparison Reference (DFIR; *Rasmussen et al.,* 2012) wind shield. As reported in (*Notaros et al.*, 2016), the 2/3-scale DFIR was effective in reducing the ambient wind speeds by nearly



a factor of 2-3 based on data from outside and inside the fence. The flow field in and
around the DFIR has been simulated by (*Theriault et al.,* 2015) assuming steady
ambient winds. They found that depending on the wind direction relative to the
octagonal fence, weak up/down drafts could be generated above the sensor areas. For
5 m/s speeds, the up/down drafts could range between -0.4 (down) to 0.2 m/s (up).
The instrument set-up was the same for the two sites (Greeley, Colorado and
Huntsville, Alabama). Huntsville has a very different climate from Greeley, and its
altitude is 212 m MSL as compared with 1.4 km MSL for Greeley. According to the
Köppen–Trewartha climate classification system (*Trewartha and Horn,* 1980), this labels
Greeley as a semiarid-type climate, whereas Huntsville is a humid subtropical-type
climate (*Belda et al.,* 2014).
The MPS is an optical array probe (OAP) that uses the technique introduced by
*Knollenberg* (1970, 1976, 1980) and measures drop diameter in the range from 0.05-3.1
mm. A 64 element photo-diode array is illuminated with a 660 nm collimated laser
beam. Droplets passing through the laser cast a shadow on the array and the decrease
in light intensity on the diodes is monitored with the signal processing electronics. A two
dimensional image is captured by recording the light level of each diode during the
period that the array is shadowed. The fall velocity is derived using two methods. One
uses the same approach as described in (*Montero-Martinez et al.,* 2009) where the fall
velocity is calculated from the product of the true air speed clock and ratio of the image
height -to-width. Note that "width" is the horizontal dimension parallel to the array and
"height" is along the vertical. The second method computes the fall velocity from the
maximum horizontal dimension (spherical drop shape assumption) divided by the
amount of time that the image is on the array, a time measured with a 2 MHz clock. In
order to be comparable to the results of (*Montero-Martinez et al.,* 2009), their approach
is implemented here for sizes > 250 µm. The fall velocity of smaller, slower moving
droplets, are measured using the second technique.
The limitations and uncertainties associated with OAP measurements have been well
documented (*Korolev et al.,* 1991; 1998; *Baumgardner et al.,* 2017). All possible
corrections have been applied, including the removal of artifacts due to splashing, and
oversizing that results from out-of-focus droplets (*Korolev* 2007). The sizing and fall
speed errors primarily depend on the digitization error (± 25 microns). The fall speed
accuracy according to the manufacturer (DMT) is <10% for 0.25 mm and <1% for sizes
greater than 1 mm, limited primarily by the accuracy in droplet sizing.

The 3[rd] generation 2DVD is described in detail by (*Schoenhuber et al.,* 2007; 2008) and
its accuracy of size and fall speed measurement has been well documented (e.g.,





*Thurai et al.,* 2007; 2009; *Huang et al.*, 2008; *Bernauer et al.*, 2015). Considering the
horizontal pixel resolution of 170 microns and other factors, the effective sizing range is
D> 0.7 mm. The fall velocity accuracy is determined primarily by the accuracy of
calibrating the distance between the two orthogonal light "sheets" or planes and is < 5%
for fall velocity <10 m s$^{-1}$. In our application, we utilize the MPS for measurement of
small drops with D < 1.2 mm and to compare the measurements with the 2DVD in the
overlap region of D ≈ 0.7–2.0 mm to ensure consistency of observations. The only fall
velocity threshold used for the 2DVD is the lower limit set at 0.5 m s$^{-1}$ in accordance
with the manufacturer guidelines for rain measurements.

## 2.1 Fall Speeds from Greeley, Colorado

We first consider a long duration (around 20 h) rain episode on 17 April 2015 which
consisted of a wide variety of rain types/rates (mostly light stratiform < 8 mm h$^{-1}$) as
described in Table 2 of (*Thurai et al.,* 2017). Two wind sensors at height of 1 m were
available to measure the winds outside and inside the DFIR. Average wind speeds
were, respectively, < 1.5 m s$^{-1}$ inside the DFIR and < 4 m s$^{-1}$ outside with light gusts.
These wind sensors were specific to the winter experiment described in (*Notaros et al.,*
2016) and were unavailable for the rain measurement campaign after May 2015.
Figure 1(a) shows the fall speeds versus D from the 2DVD (shown as contoured
frequency of occurrence), along with mean and ±1σ standard deviation from the MPS.
Also shown is the (*Atlas et al.*, 1973) fit to the terminal fall speed measurements of
(*Gunn and Kinzer*, 1949) at sea level and after applying altitude corrections (*Beard,*
1976) for the elevation of 1.4 km MSL for Greeley. Panel (b) shows the histogram of fall
speeds for two selected diameter intervals (0.5±0.1 mm) and (1±0.1 mm). Panel (a)
demonstrates the excellent "visual" agreement between the two instruments in the
overlap size range (0.7-2 mm) as well as with the fit to the Gunn-Kinzer laboratory data.
Notable is the remarkable agreement in mean fall speeds between the Gunn-Kinzer fit
and the MPS for D< 0.5 mm down to near the lower limit of the instrument (0.1 mm).
Few measurements have been reported of fall speeds in this size range.
The histograms in Fig. 1(b) show good agreement between 2DVD and MPS for 1 mm
drop sizes. The visual agreement between the two instruments is excellent with respect
to the mode, symmetry, spectral width and lack of skewness in the distributions. The
mean is 3.8 m s$^{-1}$ while the spectral width or standard deviation from MPS data is 0.6 m
s$^{-1}$. The corresponding coefficient of variation (ratio of standard deviation to mean) is
15.7%. The finite bin width (±0.1 mm) used causes a "spread" of around 0.5 m s$^{-1}$ which
is clearly a significant contributor to the measured coefficient of variation. The definition
of sub- or super-terminal fall speeds by (Montero-Martinez et al., 2009) is based on fall



speeds that are, respectively, less than 0.7 times the mean value or greater than 1.3
times the mean value (i.e., exceeding 30% threshold on either side of the mean terminal
fall speed). From examining the 1 mm size fall speed histogram there is negligible
evidence of occurrences with fall speeds <2.66 m s$^{-1}$ (sub) or >4.94 m s$^{-1}$ (super).

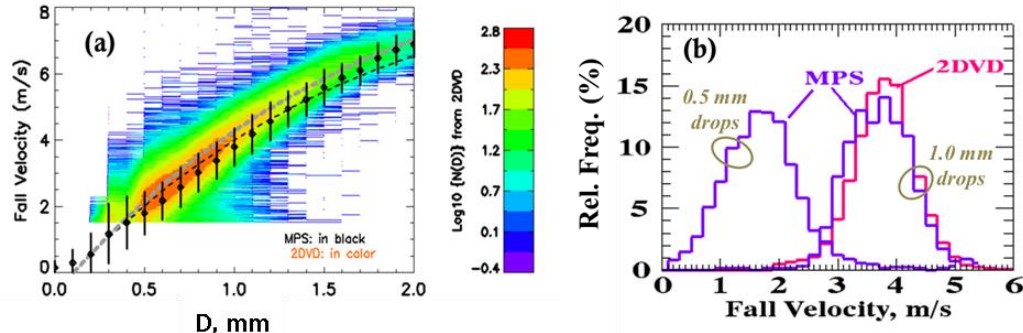


*Figure 1. (a) Fall velocity versus diameter (D). The contoured frequency of occurrence from*
*2DVD data is shown in color (log scale). The mean fall velocity and ±1σ standard deviation bars*
*are from MPS. The dark dashed line is from the fit to the laboratory data of Gunn and Kinzer*
*(1949) and the grey dashed line is the same except corrected for the altitude of Greeley, CO*
*(1.4 km MSL). (b) Relative frequency histograms of fall velocity for the 0.5±0.1 mm and 1±0.1*
*mm bins.*

The histogram from MPS for the 0.5 mm sizes shows positive skewness with mean of
1.8 m s$^{-1}$, spectral width of 0.65 m s$^{-1}$ and corresponding coefficient of variation nearly
doubling to 35%. The finite bin width (±0.1 mm) causes a "spread" of 0.4 m s$^{-1}$ which
contributes to the measured coefficient of variation. Nevertheless, it is not possible to
rule out the occurrence of sub- or super-terminal fall speeds, respectively, less than
1.26 m s$^{-1}$ or exceeding 2.34 m s$^{-1}$ (i.e., exceeding 30% of the mean value) based on
our data.

2.2 Fall Speeds from Huntsville, Alabama
The first Huntsville event occurred on 11 April 2016 and consisted of precipitation
associated with the mesoscale vortex of a developing squall line that moved across
northern Alabama between 1800 and 2300 UTC and produced over 25 mm of rainfall in
the Huntsville area. Figure 2(a) shows the ambient 10-m height wind speeds (3 s and 5-





min averaged) recorded at the site. Maximum speeds were less than 5 m s$^{-1}$ and wind
gusts were light. As no direct *in situ* measurement of turbulence was available we use
the approach by (*Garrett and Yuter,* 2014) who estimate the difference between the
maximum wind speed, or gust, that was sampled every 3 s, and the average wind
speed from successive 5 min intervals. The estimated turbulent intensity is proportional
to $E = ($Gusts$-$ AverageWind$)^2/2$. Figure 2(b) shows the $E$ values which were small
(maximum $E < 0.4$ m$^2$ s$^{-2}$) indicative of low turbulence. Also, shown in Fig. 2(b) is the
2DVD-based time series of rainfall rate (R) averaged over 3 mins; the maximum R is
around 10 mm h$^{-1}$.

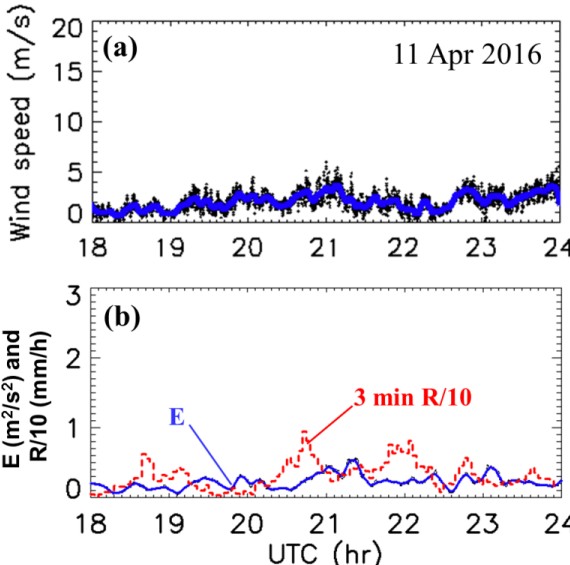


*Figure 2: (a) 3-s raw and 5-min averaged wind speeds at 10-m height. (b) turbulent*
*intensity estimates E, and 3-min averaged R (note: plot is R/10).*

Figure 3(a) shows the fall velocity versus D comparison between the two instruments
while panel (b) shows the histograms for the 0.5 and 1 mm bins. Similar to the Greeley
event, the mean fall speed agreement between both instruments in the overlap region is
excellent and consistent with the fit to the Gunn-Kinzer laboratory data. As in Fig. 1(a),
the MPS data in Fig. 3(a) is in excellent agreement with Gunn-Kinzer fit for sizes < 0.5
mm.
The 0.5 and 1 mm histogram shapes in Fig. 3(b) are quite similar to the Greeley case
shown in Fig. 1(b).  The mean and standard deviations from the MPS data for the 0.5
and 1 mm bins are, respectively, [2  0.62] and [3.88  0.44] m s$^{-1}$. The comments made



earlier with respect to Fig. 1(b) of the Greeley event are also applicable here. In
particular, the fall speed histogram for the 0.5 mm sizes cannot rule out the occurrence
of sub- or super-terminal fall speeds based on our data.

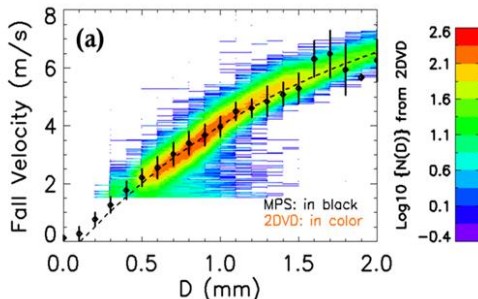
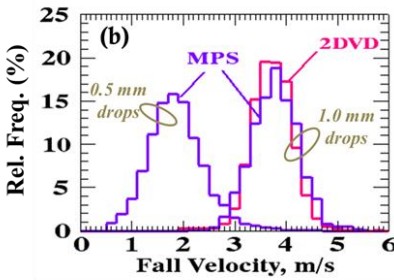


*Figure 3. (a) as in Fig. 1(a) except for 11 April 2016 event. The dashed line is fit to*
*Gunn-Kinzer at sea level. (b) as in Fig. 1(b) except for 11 April 2016 event.*

The second case considered is from 30 November 2016 wherein a supercell passed
over the instrumented site from 0300-0330 UTC producing about 15 mins later a long-
lived EF-2 tornado. Strong winds were recorded at the site with 5-min averaged speeds
reaching 10-12 m s$^{-1}$ between 0320-0330 and E values in the range to 7-8 m$^2$ s$^{-2}$
indicating strong turbulence (Fig. 4a,b). The rain rates peaked at 70 mm h$^{-1}$ during this
time (Fig. 4b). About 3 h later several squall-line type storm cells passed over the site
from 0700-0900 UTC again with strong winds but considerably lower E values 2-4 m$^2$ s$^{-2}$
and maximum R of 80 mm h$^{-1}$. After 1000 UTC the E values were much smaller (< 0.5
m$^2$ s$^{-2}$) indicating calm conditions. The peak R is also smaller at 30 mm h$^{-1}$ at 1000 UTC.
Figure 4 panels (c) and (d) show the mean and ±1σ of the fall speeds from the 2DVD for
the 1.3 and 2 mm drop sizes, respectively. The MPS data are not shown here since
during this event it was located outside the DFIR on its turntable and we did not want to
confuse the wind-effects between the two instruments. It is clear from Fig. 4(c) that
during the supercell passage (0300-0330 UTC) the mean fall speed for 1.3 mm drops
decreases (from 5 to 3.5 m s$^{-1}$) and the standard deviation increases (from 0.5 to 1.5 m
s$^{-1}$). The same trend can be seen for the subsequent squall-line rain cell passage from
0700-0900 UTC.



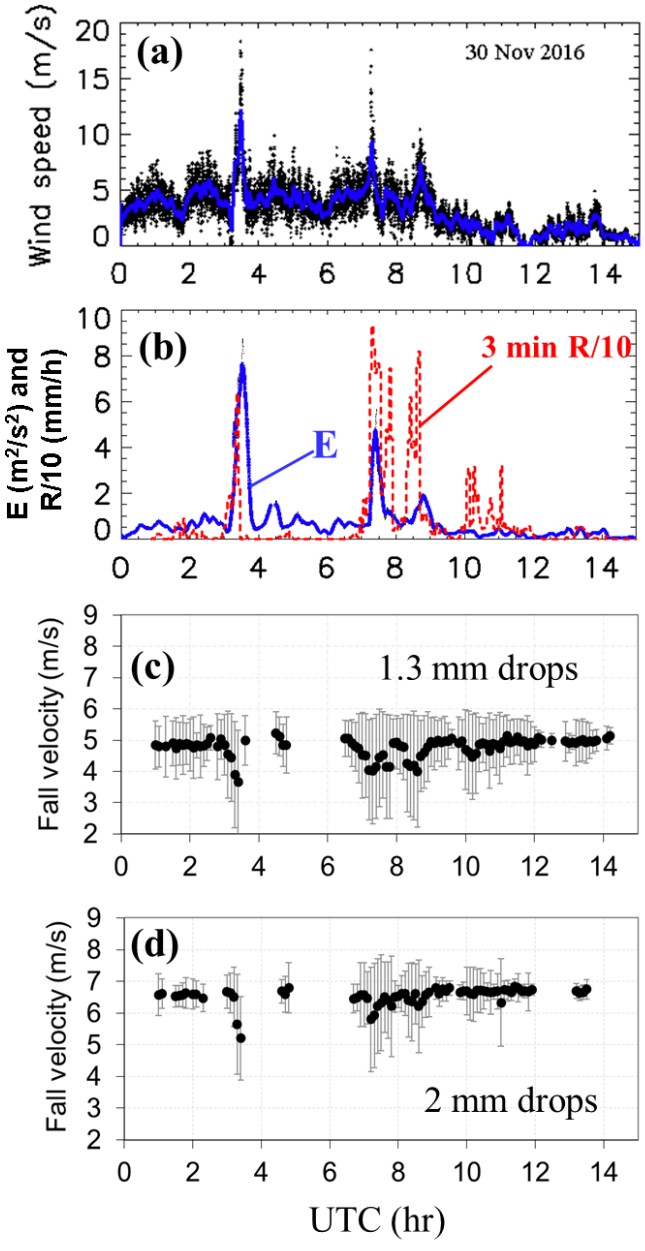


Figure 4. (a) as in Fig. 2(a) except for 30 Nov 2016 event. (b) as in Fig. 2(b). (c) mean
and ±1σ standard deviation of fall speeds from 2DVD for 1.3±0.1 mm sizes. (d) as in (c)
except for 2±0.1 mm sizes.



To expand on this observed correlation, Fig. 5 (a,b) show scatterplots of the mean fall
speed and standard deviation versus $E$ for the 1.3 mm drops (panels c and d show the
same but for the 2 mm drops). The mean fall speed decreases with increasing $E$ nearly
linearly for $E>1$ m$^2$ s$^{-2}$. This decrease relative to *Gunn-Kinzer* terminal fall speeds is
termed as "sub-terminal" and our data is in general agreement with (*Montero-Martinez*
*and Garcia-Garcia* 2016) who found an increase in the numbers of sub-terminal drops
with sizes between 1-2 mm under windy conditions using a 2D-Precipitation probe with
resolution of 200 microns (similar to 2DVD) but without wind fence. The standard
deviation of fall speeds ($\sigma_f$) versus $E$ is shown in panels 6 (b,d). When $E>1$ m$^2$ s$^{-2}$, the $\sigma_f$
is nearly constant at 1.5 m s$^{-1}$ for both drop sizes. For $E<1$, the $\sigma_f$ is more variable and
essentially uncorrelated with $E$. From the discussion related to Fig. 1(b) and 3(b), $\sigma_f$
values exceeding 0.5 m s$^{-1}$ can be attributed to physical as opposed to instrumental and
finite bin width effects. Thus, the fall speed distributions are considerably broadened
when $E>1$ m$^2$ s$^{-2}$ due to increasing turbulence levels which is again consistent with the
findings of (*Montero-Martinez and Garcia-Garcia*, 2016) as well as (*Garett and Yuter*,
2014). The latter observations, however, were of graupel fall speeds in winter
precipitation using a multi-angle snowflake camera (*Garrett et al.,* 2012).

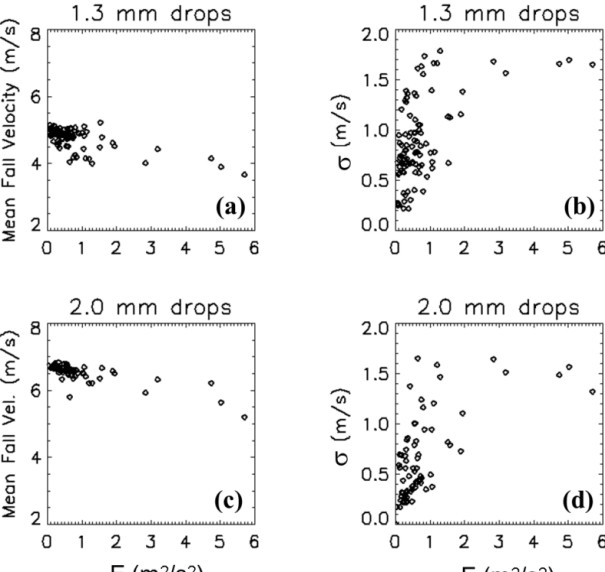


*Figure 5. (a,b) mean fall speed and standard deviation, respectively, versus E for 1.3*
*mm sizes. (c,d) same but for 2 mm sizes.*



## 3 Discussion and Conclusions

We have reported on raindrop fall speed distributions using a high resolution (50 microns) droplet spectrometer (MPS) collocated with moderate resolution (170 microns) 2DVD to cover the entire size range (0.1 mm onwards) expected in natural rain. The only comparable earlier study is by (*Montero-Martinez et al.*, 2009) who used collocated 2D-cloud and precipitation probes (2D-C, 2D-P) but restricted their data to calm wind conditions. Their main conclusion was that the distribution of the ratio of the measured fall speed to the terminal fall speed for 0.44 mm size, while having a mode at 1 was strongly positively skewed with tails extending to 5 especially at high rain rates. In our data shown in Fig. 1(b) and 3(b), there is no such strong positive skewness, and the corresponding ratio does not exceed 2.

*Larsen et al.,* (2014) appear to confirm the ubiquitous existence of super-terminal fall speeds for sizes < 1 mm using different instruments one of them being a 2DVD. It is well-known that "mis-matched" drops cause erroneous fall speed estimates from 2DVD for tiny drops. To clarify the "mis-matched" drop problem: it is very difficult to match a drop detected in the top light-beam plane of the 2DVD to the corresponding drop in the bottom plane for sizes < 0.5 mm (*Schoenhuber et al.,* 2008; Appendix in *Huang et al.*, 2010; *Bernauer et al.*, 2015). It is not clear if (*Larsen et al.*, 2014) accounted for this problem in their analysis.

Our histograms of fall speeds for 1 mm sizes (Fig. 1b and 3b) under calm wind conditions from both MPS and 2DVD did not show any evidence of either sub- or super-terminal speeds, rather the histograms were symmetric with mean close to the Gunn-Kinzer terminal velocity value. However, for the 0.5 mm sizes, our histogram of fall speeds using the MPS under calm conditions cannot rule out the occurrence of both sub- and super-terminal fall speeds, after accounting for instrumental and finite bin width effects.

In a later study using only the 2D-P probe, (*Montero-Martinez and Garcia-Garcia*, 2016) found sub-terminal fall speeds and broadened distributions under windy conditions for 1-2 mm sizes in general agreement with our results using 2DVD. *Stout et al.*, (1995) simulated the motion of drops in isotropic turbulence and determined that there would be a significant reduction of the average drop settling velocity (relative to terminal velocity) of greater that 35% for drops around 2 mm size when the ratio of *rms* velocity fluctuations (due to turbulence) relative to drop terminal velocity is around 0.8. Whereas we did not have a direct measure of the *rms* velocity fluctuations, the proxy for turbulence intensity ($E$) related to wind gusts during supercell passage (very large $E$ around 7 m$^2$ s$^{-2}$) clearly shows a significant reduction in mean fall speeds of 25-30% relative to terminal speed for 1.3 and 2 mm sizes with significant broadening of the fall speed distributions relative to calm conditions.





When $E$<0.5-1 m$^2$ s$^{-2}$, our data show that the mean fall speeds are within a few per cent
(<5%) of the *Gunn-Kinzer* terminal velocity over the entire range from 100 microns and
larger to precipitation sizes.  For sizes > 1 mm, no significant broadening of the fall
speed distribution over that ascribed to instrument and/or finite bin widths effects were
observed. While our dataset is limited to three events they cover a wind range of rain
rates, wind conditions and two different climatologies. Analysis of further events with
direct measurement of turbulent intensity would be needed to generalize our findings.

## Data Availability

Data used in this paper can be accessed from:
ftp://lab.chill.colostate.edu/pub/kennedy/merhala/Bringi_et_al_2017_GRL_datasets/

## Competing interests

VNB and MT declare they have no conflict of interest. DB is employed by Droplet
Measurements Technologies, Inc. located in Longmont, Colorado, USA who make the
Meteorological Particle Spectrometer used in this study.

## Acknowledgements

Two of the authors (VNB and MT) acknowledge support from the U.S. National Science
Foundation via grant AGS-1431127. The assistance of Dr. Patrick Gatlin of
NASA/MSFC is gratefully acknowledged. Prof. Kevin Knupp and Mr. Carter Hulsey of
the University of Alabama in Huntsville processed the wind data.






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
