# Peer review of "Raindrop Fall Velocities from an Optical Array Probe and 2D-Video Disdrometer"

_Atmospheric Measurement Techniques, 2017_

## Referee Comment (RC1) · Anonymous Referee #1 · 11 Dec 2017

**REVIEW REPORT**

Review of amt-2017-401

By Viswanathan Bringi, Merhala Thurai and Darrel Baumgardner

Manuscript Title – Raindrop Fall Velocities from an Optical Array Probe and 2D-Video Disdrometer

**GENERAL COMMENTS**

In the manuscript the Authors analyzed three precipitation events occurred in USA (Colorado and Alabama) that differs for the climatology of the colocations and for the wind conditions. The aim of the study is to evaluate the wind influence on the raindrops terminal fall speed measured by two different type of devices, namely the Meteorological Particle Spectrometer (MPS) and the 2D video disdrometer (2DVD). The manuscript is well organized however I think that the aim of the study and in particular its practical applications should be specified in the Introduction section. As stated also by the Authors more case studies should be added or at least the analysis should be extended to larger drops (see for example comment 6 and 9 below). Furthermore section 2 need to be enlarged with information regarding the data processing (see comment 3 below) and more analytical comparison should be done to confirm the consistency of fall speed measurements from the two devices (see comment 1 below). Finally I have some specific comments, that are shown below.

**SPECIFIC COMMENTS**

1.      Line 24-25: in the manuscript the consistency of fall speed measurements from the two devices is provided only qualitatively (i.e. "excellent visual agreement") some quantitative results should be provided for all the diameters in the overlapping region.
2.      Line 109: please clarify which are the "other factors" that gives the threshold of 0.7 mm for the drop diameter. The 2DVD is able to measures drops with D < 0.7 mm. Usually the minimum detectable diameter for 2DVD is considered 0.2 mm or 0.3 mm. In this case the overlapping between the two instruments can be enlarged. Please provide a clarification of this threshold or consider the option of enlarging the overlapping region.
3.      Line 114-115: As reported in numerous papers in the literature, the 2DVD measures a number of spurious drops that can are usually removed from the data using proper filter criterion, such as the one based on the relation between measured and theoretical fall velocities. Please note that in my experience most of the spurious drops have small diameters (D< 2 mm) and therefore are within the range of diameters analyzed in this study.  Did the Authors use any kind of criterion to filter out these drops? If yes which is the impact of the filtering on the results. If not, how can the Authors be sure that those drops are real drops and not spurious ones? I think that the Authors should clarify this point in the manuscript because it is crucial for the validity on the results obtained in the study.
4.      Line 119: How do the Authors identify the different rain types?
5.      Line 131: I suggest to change the word "excellent" with the word "good". The MPS underestimates the fall velocities for 0.7 mm < D < 1 mm with respect to 2DVD, while the 2DVD

overestimates the fall velocities for 1 mm < D < 2 mm with respect to the Gunn and Kinzer fit. Furthermore a more quantitative agreement should be performed.

6.      Figure 1b and Figure 2b: I suggest to plot the fall velocity histogram also for other drop diameters (let say 0.7 mm and 1.5 mm for example) so the readers can have more cases to evaluate the agreements between 2DVD and MPS.

7.      Line 187: similarly to comment 5, also here the word "excellent" is not appropriate due to the overestimation of MPS with respect to Gunn and Kinzer fit for D < 0.5 mm.

8.      Figure 3a: can the Authors provide an explanation of the differences in the mean fall velocity between Gunn and Kinzer fit and MPS measurements for D > 1.5 mm?

9.      Figure 5: what about large drops? Which is the effect of wind on large drops? I suggest to use the 2DVD data to made the same analysis for larger D.

**TECHNICAL CORRECTIONS**

1.      Line 286: probably "wind range" should be "wide range".

---

## Referee Comment (RC2) · H. Leijnse (Referee) · 11 Dec 2017

This paper describes results from two measurement campaigns with a Meteorological Particle Sensor (MPS) and a 2D-Video Disdrometer (2DVD). The analyses presented in this paper are focussed on the fall speeds of droplets measured by the different instruments, and whether these deviate from results from laboratory experiments (super- or sub-terminal fall speeds). Observed sub-terminal fall speeds are then linked to turbulence intensity. I think this is an interesting paper. It contributes to the scientific discussion on the puzzling super-terminal small raindrops by showing results where these were not observed. However, the paper would benefit from a clearer description of its aims, and, if possible, stronger conclusions. As far as I understand it, there are three main messages in the paper: 1) there is no evidence of super-terminal raindrops

(contrary to Montero-Martinez et al., 2009 and Larsen et al., 2014); 2) the fall velocities of real drops closely follow the relations found by Gunn and Kinzer (1949) down to very small drops; and 3) there is a clear effect of strong turbulence on the mean and the standard deviation of fall speeds of drops of a given diameter. If this is indeed the case, then I think this should be more clearly stated in the introduction, and should be discussed more elaborately in the conclusions. So I think that after revisions, this paper is suitable for publication in *Atmospheric Measurement Techniques*. Specific comments are given below.

**Specific comments**

1. In the introduction it should be more clearly stated what the exact aims of this paper are.

2. On lines 66-74, the use of a DFIR is discussed along with its effects on the local wind field. For studying the relation between turbulence intensity and raindrop fall speeds, how does this double fence affect the turbulence just above the instrument? I can imagine that by reducing the average wind speed, the turbulence is also reduced. On the other hand, as stated on line 74, the fence itself also generates up- and downdrafts. I think that the effects of the use of a DFIR on the results presented in this paper should be discussed and, if possible, quantified.

3. In Fig. 1 (especially panel a) the MPS seems to detect slightly (but systematically) lower fall velocities than the 2DVD in the Greeley data. This is not the case for the Huntsville data (Fig. 3). Please give an explanation for this.

4. On lines 208-215, the correlation between $E$ on the one hand, and the mean and standard deviation of the raindrop fall speeds on the other is discussed. I agree that this correlation is there. However, judging from Fig. 4, I think there is also
some correlation with the rain rate $R$ (especially the peak at 10 UTC). Please elaborate on the role of the rain rate for these correlations.

5. On lines 224-225, the observed near-linear decrease of the mean fall speed with turbulence intensity (or at least its proxy $E$) is mentioned. How significant is this relation?

6. On lines 247-254, the results presented by Montero-Martinez et al. (2009) are compared to those presented in this paper. What could be the explanation for this difference? Could it be something similar to what is discussed in the next paragraph (lines 255-262) about the findings of Larsen et al. (2014)? Please elaborate on this.

7. On lines 272-281, the relation to the findings of Stout et al. (1995) are discussed. Is there an empirical relation between $E$ and the rms velocity fluctuations due to turbulence? If so it would be interesting to see whether the 35% reduction in mean velocity is observed at similar rms velocity fluctuations-to-terminal fall speed ratios (0.8).

**Technical comments**

1. In Figs 2b and 4b, would it be possible to use a second $y$-axis for $R$ instead of presenting $R/10$ on the existing $y$-axis?

2. On line 275, "greater that" should be "greater than".

---

## Referee Comment (RC3) · Anonymous Referee #3 · 12 Dec 2017

Review of AMT-2017-401
By V. Bringi, M. Thurai and D. Baumgardner
Manuscript Title – Raindrop Fall Velocities from an Optical Array Probe and 2D-Video Disdrometer.

This manuscript reports on raindrop fall velocity measurements by using two different instruments: a MPS (Meteorological Particle Spectrometer) which measures drops in the 0.1-3 mm range, and the wildly used 2DVD (two-Dimensional Video Disdrometer), which measures size and fall velocity of drops between 0 and 10 mm. The MPS and 2DDVD were used to measure fall velocity of drops in the 0.1-2 and larger than 0.7 mm diameter range. The overlapping region 0.7-2 mm diameter was used to cross-validate the two measurements.

Three different case studies were analyzed in order to relate the properties of the drop fall velocity to different precipitation systems (one stratiform, one squall line and one super-cell case with low and high turbulence associate for the first two and the third case, respectively).

The paper is linear and quite easy to read. I have only one major comment that can give a contribution, in my opinion, to the generalization of the results. It is reported below together with minor comments that, once addressed, will allow the publication of the paper on the Atmospheric Measurement Techniques journal.

Major comment.

- Section 2.2: in the Section 2.1 the authors investigated a stratiform case, while in the Section 2.2 a squall line and a super-cell case. The squall line case reported generally low rainfall rate and turbulence (comparable to the values registered in the stratiform case). It could be useful, in my opinion add (or substitute) a convective event, a sort a middle point between a convective and tornadic case, in order to have a general overview of the characteristics of drop fall velocity in a broader range of precipitation systems.

Minor comments.

- Line 141: what does it mean that the finite bin width causes a spread of 0.5 m/s? Can the authors explain better? The same is reported in other parts of the text.

- Lines 211-215: what is the explanation that the authors give to the decrease of fall speed during the most intense wind and rainfall rate? Does it can be related to the presence of ascending flow?

- Lines 224-225: similar to the previous comment. How do they justify the decrease of fall speed when $E$ (turbulence) increases?

- Panel (b) of Figures 2 and 4: the rain rate should be reported on the right y-axis avoiding the necessity to show its values scaled on a factor ten.

- Figure 2a: the y-axis limit should not exceed 10 m/s to improve the detail of the plot.

---

## Referee Comment (RC4) · Anonymous Referee #4 · 18 Dec 2017

1. Does the paper address relevant scientific questions within the scope of AMT? YES

2. Does the paper present novel concepts, ideas, tools, or data? YES

3. Are substantial conclusions reached? YES

4. Are the scientific methods and assumptions valid and clearly outlined? YES

5. Are the results sufficient to support the interpretations and conclusions? YES

6. Is the description of experiments and calculations sufficiently complete and precise to allow their reproduction by fellow scientists (traceability of results)? YES, with a few additions requested in below remarks.

7. Do the authors give proper credit to related work and clearly indicate their own

[Figure]

new/original contribution? YES

8. Does the title clearly reflect the contents of the paper? YES

9. Does the abstract provide a concise and complete summary? YES

10. Is the overall presentation well structured and clear? YES

11. Is the language fluent and precise? YES

12. Are mathematical formulae, symbols, abbreviations, and units correctly defined and used? YES

13. Should any parts of the paper (text, formulae, figures, tables) be clarified, reduced, combined, or eliminated? Minor, as requested in below comments

14. Are the number and quality of references appropriate? YES

15. Is the amount and quality of supplementary material appropriate? YES

==========================

GERNERAL COMMENTS

This is an excellent paper, addressing the issue of in-situ measurements of rain drops over their full size range, with focus on the drops' fall velocities. Especially the characteristics of very small drops still are not investigated thoroughly up to here. The investigation of the relationship between wind turbulence and rain drops' fall velocity is convincing and in the future will allow new refinements in the related remote sensing algorithms. The study is presented in a clear and concise way. The reviewer likes to point out, that such work does not only address academic interests, but points to a number of applications, like remote sensing of the atmosphere, as the authors shortly mention in their introduction. Other applications include knowledge of channel characteristics for satellite communications at EHF frequency bands. Thus the study is very relevant for these fields of science and applications. The work is based on the combined analyses of data from 2 instruments at same site, the Meteorological Particle Spectrometer (MPS) and the 2D-video disdrometer (2DVD). Field measurement of rain drops still may be considered as challenge, with quite a few solutions being around, but none of the instruments may claim to be perfect. The presented results do rise a few relevant questions, which are given in below specific comments. Summarizing it is said, that this paper gives an excellent contribution, bringing new aspects for the relevant fields of science and applications.

SPECIFIC COMMENTS

*** Line 98 / 99: "All possible corrections have been applied, including the removal of artifacts due to splashing,..." Careful preprocessing, verification and validation of data is of utmost importance for the present study. The results presented require correctness of challenging measurement processes. Some of the results given actually leave a few questions open, as addressed in below comment to Fig 1a and Fig 3a., To be sufficiently self-contained, thus the paper might shortly describe the mentioned correction algorithms.

*** Line 157: "The histogram from MPS for the 0.5 mm sizes shows positive skewness" Can the authors give an explanation / discussion / assumption?

*** Fig. 1a and Fig. 3a: Mean fall velocities from the MPS are read from these figures approximately as:

Fig. 1a: D = 0,7 mm, v = 2.55 m/s

Fig. 3a: D = 0.7 mm, v = 3.05 m/s

That represents an exceedance by more than 16 % (Fig. 3a over Fig. 1a), in spite of the lower pressure in Greely (Fig 1a) leading to the expectation of faster drops than in Huntsville (Fig 3a). The authors please could discuss this.

*** Fig. 3a: Mean fall velocities from the MPS are read from this figure approximately as:

[Figure]

D = 1.5 mm, v = 5.28 m/s

D = 1.6 mm, v = 6.31 m/s

D = 1.7 mm, v = 6.5 m/s

D = 1.8 mm, v = 5.93 m/s

These values differ significantly from the fit to Gunn-Kinzer, further from the expected monotonic behaviour. The authors please could discuss this.

TECHNICAL COMMENTS:

\*\*\* Line 230: "($\sigma$f) versus E is shown in panels 6 (b,d)." It probabaly should read as "($\sigma$f) versus E is shown in panels (b,d) of Fig 5."

---

## Author Comment (AC1) · 21 Dec 2017

To:

Editor Dr Gianfranco Vulpiani:

We respond to the 4 reviewer's comments in sequence from RC1 to RC4. Our response is in italics and the text modifications are highlighted. At the end of our Response we attach the revised manuscript where changes are highlighted.

Please let me know if you need further information.

VN Bringi

[Figure]

Please also note the supplement to this comment:
https://www.atmos-meas-tech-discuss.net/amt-2017-401/amt-2017-401-AC1-supplement.pdf

[Figure]

**Supplement:**

| 1                                | REVIEW REPORT                                                                                                                                                                                                                                                                                                                                                                                                                                                                                                                                                                           |
|----------------------------------|-----------------------------------------------------------------------------------------------------------------------------------------------------------------------------------------------------------------------------------------------------------------------------------------------------------------------------------------------------------------------------------------------------------------------------------------------------------------------------------------------------------------------------------------------------------------------------------------|
| 2                                |                                                                                                                                                                                                                                                                                                                                                                                                                                                                                                                                                                                         |
| 3                                | Review of amt-2017-401 by RC1                                                                                                                                                                                                                                                                                                                                                                                                                                                                                                                                                           |
| 4                                |                                                                                                                                                                                                                                                                                                                                                                                                                                                                                                                                                                                         |
| 5                                | By Viswanathan Bringi, Merhala Thurai and Darrel Baumgardner                                                                                                                                                                                                                                                                                                                                                                                                                                                                                                                            |
| 6                                |                                                                                                                                                                                                                                                                                                                                                                                                                                                                                                                                                                                         |
| 7                                | Manuscript Title – Raindrop Fall Velocities from an Optical Array Probe and 2D-Video Disdrometer                                                                                                                                                                                                                                                                                                                                                                                                                                                                                        |
| 8                                |                                                                                                                                                                                                                                                                                                                                                                                                                                                                                                                                                                                         |
| 9                                |                                                                                                                                                                                                                                                                                                                                                                                                                                                                                                                                                                                         |
| 10                               | GENERAL COMMENTS                                                                                                                                                                                                                                                                                                                                                                                                                                                                                                                                                                        |
| 11                               |                                                                                                                                                                                                                                                                                                                                                                                                                                                                                                                                                                                         |
| 12
13
14
15             | In the manuscript the Authors analyzed three precipitation events occurred in USA (Colorado and Alabama) that differs for the climatology of the colocations and for the wind conditions. The aim of the study is to evaluate the wind influence on the raindrops terminal fall speed measured by two different type of devices, namely the Meteorological Particle Spectrometer (MPS) and the 2D video disdrometer                                                                                                                                                                     |
| 16                               | (2DVD). The manuscript is well organized however I think that the aim of the study and in particular its                                                                                                                                                                                                                                                                                                                                                                                                                                                                                |
| 17
18
19
20
21
22 | practical applications should be specified in the Introduction section. As stated also by the Authors more
case studies should be added or at least the analysis should be extended to larger drops (see for
example comment 6 and 9 below). Furthermore section 2 need to be enlarged with information
regarding the data processing (see comment 3 below) and more analytical comparison should be done
to confirm the consistency of fall speed measurements from the two devices (see comment 1 below).
Finally I have some specific comments, that are shown below. |
| 23
24                         | We appreciate the reviewer's comments and our response is given in italics while the modifications to the manuscript are highlighted as below.                                                                                                                                                                                                                                                                                                                                                                                                                                          |

25 General: The first sentence in the Introduction gives three applications with three pertinent 26 references. We believe these are also "practical" references. For example, the modeling of

27 collisional processes is in part based on assigning a unique terminal fall speed for a given mass

of raindrop. Same is true for retrieval of DSD from vertical pointing profilers. We have also

added in a new reference to Yu et al. (2016) in AMT who also suggest that ambient flow and

30 turbulence may play a role in modifying drop fall speeds.

31 We have added the following sentences to clarify this point in the Introduction:

| 32       | <mark>"In the</mark>                                                                     | ese and other applications it is nearly universally accepted that there is a unique                      |  |
|----------|------------------------------------------------------------------------------------------|----------------------------------------------------------------------------------------------------------|--|
| 33       | fall speed ascribed to drops of a given mass or diameter and that it equals the terminal |                                                                                                          |  |
| 34       | speed with adjustment for pressure (e.g., Beard 1976). "                          |                                                                                                          |  |
| 35       | " <mark>Thus</mark>                                                                      | , there is some evidence that rain drops may not fall at their terminal velocity                         |  |
| 36       | <mark>excep</mark>                                                                       | t under calm conditions and that the concept of a fall speed distribution for a drop                     |  |
| 37       | of given mass (or, diameter) might need to be considered which is the topic of this      |                                                                                                          |  |
| 38       | paper. The implications are rather profound especially for numerical modeling of         |                                                                                                          |  |
| 39       | collision-coalescence and breakup processes which are important for shaping the drop     |                                                                                                          |  |
| 40       | <mark>size d</mark>                                                                      | listribution."                                                                                           |  |
| 41       |                                                                                          |                                                                                                          |  |
| 42       | 2 Our response to SPECIFIC COMMENTS.                                                     |                                                                                                          |  |
| 43       | SPECIF                                                                                   | ICCOMMENTS                                                                                               |  |
| 44       | 1.                                                                                       | Line 24-25: in the manuscript the consistency of fall speed measurements from the two devices            |  |
| 45       |                                                                                          | is provided only $qualitatively(i.e.$ "excellent $visual$ agreement") some $quantitative$ results should |  |
| 46       |                                                                                          | be provided for all the diameters in the overlapping region.                                             |  |
| 47       |                                                                                          |                                                                                                          |  |
| 48       |                                                                                          | We have added a new Table 1 that compares the mean and standard deviation from MPS and                   |  |
| 49
50 |                                                                                          | 2DVD for both sites in the overlap diameter range 0.7 to 2 mm.                                           |  |
| 50
51 | 2                                                                                        | Line 109: please clarify which are the "other factors" that gives the threshold of 0.7 mm for the        |  |
| 52       | ۷.                                                                                       | dron diameter. The 2DVD is able to measures drons with $D
| 62       |                                                                                          | heve added the "mis-match" problem in original line 100                                                  |  |
| 63       |                                                                                          | have added the mis-match problem in original line 109.                                                   |  |
| 64       |                                                                                          | "Considering the horizontal pixel resolution of 170 um and other factors (such as                        |  |
| 65       |                                                                                          | "mis-matched" drops), the effective sizing range is D> 0.7 mm. To clarify the                            |  |
| 66       |                                                                                          | "mis-matched" drop problem: it is very difficult to match a drop detected in the top                     |  |
| 67       |                                                                                          | light-beam plane of the 2DVD to the corresponding drop in the bottom plane for                           |  |
| 68       |                                                                                          | tiny drops resulting in erroneous fall speeds."                                                          |  |
| 69       |                                                                                          |                                                                                                          |  |

70 The smallest calibration spheres that are provided by the manufacturer is 0.5 mm and 71 these are extremely difficult to drop in the sensor area and to collect them below. 72 According to Bernauer et al. accurate sizing with errors <5% is only possible for D>1 73 mm based on solid (snow) precipitation. In our experience for rain drops which are smooth shaped the threshold is lower at 0.7 mm which is what we have quoted in our 74 paper. While the mid-point of the first "bin" of the 2DVD sizing is 0.25 mm, the accuracy 75 76 is very questionable as calibration is not possible. Hence we cannot enlarge the 77 overlapping region. The whole point of our paper is to use high resolution MPS and lower resolution 2DVD to cover the entire size range with good-to-excellent accuracy. 78

- 80 3. Line 114-115: As reported in numerous papers in the literature, the 2DVD measures a number of spurious drops that can are usually removed from the data using proper filter criterion, such as 81 82 the one based on the relation between measured and theoretical fall velocities. Please note that 83 in my experience most of the spurious drops have small diameters (D<2 mm) and therefore are 84 within the range of diameters analyzed in this study. Did the Authors use any kind of criterion to 85 filter out these drops? If yes which is the impact of the filtering on the results. If not, how can the Authors be sure that those drops are real drops and not spurious ones? I think that the 86 Authors should clarify this point in the manuscript because it is crucial for the validity on the 87 88 results obtained in the study.
- 90 We clearly state in original lines 114-115 that ..." The only fall velocity threshold used for the 2DVD is the lower limit set at 0.5 m s-1 in accordance with the manufacturer 91 quidelines for rain measurements." We do not use any velocity filter in our analysis as 92 93 doing so would be counter to the detection of sub- or super terminal fall speeds. Please see our response above to point (2) specifically the threshold of 0.7 mm which we use to 94 95 eliminate mis-matched or "spurious" drops. Our confidence of sizing and fall speed measurement using 2DVD for D>0.7 mm is reinforced by the agreement with MPS in the 96 97 overlap region (please see new Table 1). Finally, the use of a DFIR wind shield appears 98 to have reduced the occurrence of "spurious" drops.
- 99 100

108

79

- 101 4. Line 119: How do the Authors identify the different rain types?
- 102 The identification of rain types was based on CSU-CHILL radar data as described in the 103 quoted reference (Thurai et al. 2017).
- Line 131: I suggest to change the word "excellent" with the word "good". The MPS underestimates the fall velocities for 0.7 mm < D < 1 mm with respect to 2DVD, while the 2DVD overestimates the fall velocities for 1 mm < D < 2 mm with respect to the Gunn and Kinzer fit.</li>
   Furthermore a more quantitative agreement should be performed.
- 109Please see new Table 1 for a quantitative comparison between MPS and 2DVD in the110overlap region. The agreement between the two instruments is, in our opinion, excellent

for both sites given the guoted accuracies in fall speed measurement for both 111 112 instruments. 113 We have replaced the fit to Gunn-Kinzer data with the 9th order polynomial fit given in 114 Foote and du Toit (1969) as opposed to using the exponential fit of Atlas et al. (1973) in 115 both figs. 1a and 3a. The latter fit is not as accurate for the small drop end (e.g., it does 116 117 not pass through the origin). We have added in Section 2.1: 118 "Also shown is the (*Foote and du Toit 1969*) (henceforth FT fit) to the terminal fall 119 speed measurements of (Gunn and Kinzer, 1949) at sea level and after applying 120 altitude corrections (Beard, 1976) for the elevation of 1.4 km MSL for Greeley." 121 122 The pressure adjustment to the Gunn-Kinzer fit to account for the altitude (1.4 km MSL) 123 of the Greeley site is in excellent agreement with the 2DVD measurements. The slight 124 125 underestimation of the MPS fall speeds relative to the fit of Foote and du Toit to the data 126 of Gunn-Kinzer in the range 0.7-1.5 mm (max deviation of 0.5 m/s or 10%) is puzzling given the excellent agreement for the Huntsville site (sea level). We have added 127 128 sentence below. 129 "However, the altitude-adjusted FT fit is slightly higher than the measured values 130 as shown in Table 1." 131 132 133 6. Figure 1b and Figure 2b: I suggest to plot the fall velocity histogram also for other drop 134 diameters (let say 0.7 mm and 1.5 mm for example) so the readers can have more cases to evaluate the agreements between 2DVD and MPS. 135 136 137 Done as suggested...see new panel c in figs. 1 and 3. 138 7. Line 187: similarly to comment 5, also here the word "excellent" is not appropriate due to the 139 140 overestimation of MPS with respect to Gunn and Kinzer fit for D < 0.5 mm. As noted in our response to (5) above, after replacing the Atlas et al fit by the Foote and 141 du Toit fit in Fig. 3a the agreement between MPS and the latter fit is considered to be 142 excellent. 143 144 8. Figure 3a: can the Authors provide an explanation of the differences in the mean fall velocity between Gunn and Kinzer fit and MPS measurements for D > 1.5 mm? 145 146 Please see our response to (7) above. The differences in Fig. 3a are no longer an issue. 147 148 149 9. Figure 5: what about large drops? Which is the effect of wind on large drops? I suggest to use 150 the 2DVD data to made the same analysis for larger D. 151

| 152
153
154
155
156 | We have added new panels Fig. 5e,f which show the effect of wind/gusts for 3 mm drops. The trend is similar to 2 mm drops. We cannot show larger drops due to very low number of samples. |
|---------------------------------|-------------------------------------------------------------------------------------------------------------------------------------------------------------------------------------------|
| 157                             | TECHNICAL CORRECTIONS 1. Line 286: probably "wind range" should be "wide range".                                                                                                          |
| 158                             | Corrected. Thank you for pointing this out.                                                                                                                                               |
| 159                             |                                                                                                                                                                                           |
| 160                             |                                                                                                                                                                                           |
| 161                             |                                                                                                                                                                                           |
| 162                             |                                                                                                                                                                                           |
| 163                             |                                                                                                                                                                                           |
| 164                             |                                                                                                                                                                                           |
| 165                             |                                                                                                                                                                                           |
| 166                             |                                                                                                                                                                                           |
| 167                             |                                                                                                                                                                                           |
| 168                             |                                                                                                                                                                                           |
| 169                             |                                                                                                                                                                                           |
| 170                             |                                                                                                                                                                                           |
| 171                             |                                                                                                                                                                                           |
| 172                             |                                                                                                                                                                                           |
| 173                             |                                                                                                                                                                                           |
| 174                             |                                                                                                                                                                                           |
| 175                             |                                                                                                                                                                                           |
| 176                             |                                                                                                                                                                                           |
| 177                             |                                                                                                                                                                                           |
| 178                             |                                                                                                                                                                                           |

- 179 Interactive comment on "Raindrop Fall Velocities from an Optical Array Probe and 2D-video
- 180 Disdrometer" by Viswanathan Bringi et al.
- 181 H. Leijnse (Referee) hidde.leijnse@knmi.nl Received and published: 11 December 2017: RC2

182 This paper describes results from two measurement campaigns with a Meteorological Particle Sensor 183 (MPS) and a 2D-Video Disdrometer (2DVD). The analyses presented in this paper are focussed on the fall 184 speeds of droplets measured by the different instruments, and whether these deviate from results from laboratory experiments (superor sub-terminal fall speeds). Observed sub-terminal fall speeds are then 185 186 linked to turbulence intensity. I think this is an interesting paper. It contributes to the scientific 187 discussion on the puzzling super-terminal small raindrops by showing results where these were not 188 observed. However, the paper would benefit from a clearer description of its aims, and, if possible, 189 stronger conclusions. As far as I understand it, there are three main messages in the paper: 1) there is 190 no evidence of super-terminal raindrops (contrary to Montero-Martinez et al., 2009 and Larsen et al., 191 2014); 2) the fall velocities of real drops closely follow the relations found by Gunn and Kinzer (1949) 192 down to very small drops; and 3) there is a clear effect of strong turbulence on the mean and the 193 standard deviation of fallspeeds of drops of a given diameter. If this is indeed the case, then I think this 194 should be more clearly stated in the introduction, and should be discussed more elaborately in the 195 conclusions. So I think that after revisions, this paper is suitable for publication in Atmospheric

196 Measurement Techniques. Specific comments are given below.

**We thank the reviewer for the positive comments. Our response is in italics while the text modifications are highlighted and in quotes.**

199 We have added in the Introduction the following:

**200 "In these and other applications it is generally accepted that there is a unique fall speed**

- 201 ascribed to drops of a given mass or diameter and that it equals the terminal speed 202 with adjustment for pressure (e.g., *Beard* 1976)."
- 203 We are reluctant to draw firmer or more elaborate conclusions than what is stated in our paper
- since, (a) the database is rather small (3 cases but from two different climatologies), (b) we do
- not have a direct measurement of turbulence and (c) we cannot quantify if the DFIR wind shield
- is affecting the sensor area in some subtle way. These questions will be addressed in the future.
- 207 In Section 3 last para we have clarified as follows:
- 208 "One caveat is that the response of the DFIR wind shield to ambient winds in terms of producing
- 209 subtle vertical air motions near the sensor area is yet to be evaluated as future work. Analysis
- 210 of further events with direct measurement of turbulent intensity, for example using a 3D-sonic
- 211 anemometer at the height of the sensor, would be needed to generalize our findings."
- 212 Specific comments
- 1. In the introduction it should be more clearly stated what the exact aims of this paper are.

215 We have added the following in the Introduction:

216

229

240

244

248

214

217 "Thus, there is some evidence that rain drops may not fall at their terminal velocity 218 except under calm conditions and that the concept of a fall speed distribution for a drop 219 of given mass (or, diameter) might need to be considered which is the topic of this 220 paper. The implications are rather profound especially for numerical modeling of 221 collision-coalescence and breakup processes which are important for shaping the drop 222 size distribution."

- On lines 66-74, the use of a DFIR is discussed along with its effects on the local windfield. For
   studying the relation between turbulence intensity and raindrop fall speeds, how does this
   double fence affect the turbulence just above the instrument? I can imagine that by reducing
   the average wind speed, the turbulence is also reduced. On the other hand, as stated on line 74,
   the fence itself also generates up- and downdrafts. I think that the effects of the use of a DFIR
   on the results presented in this paper should be discussed and, if possible, quantified.
- As mentioned in the text we do not have a direct measurement of turbulence at the 230 height of the sensor. Rather we use the wind/gusts from the anemometer at 10 m high 231 tower as a proxy for turbulence. Regarding the DFIR perturbing our results, there is 232 unfortunately not many articles describing the DFIR affect on the ambient flow other than 233 234 the quoted reference of Theriault et al. (2015). Most articles are related to the "catch efficiency" of snow gages located inside the DFIR relative to gages with standard wind 235 236 skirts. In the future we will locate one 2DVD inside the DFIR and one outside as well as 237 collocated 3D-sonic anemometers both inside and outside...but this is another field 238 project for which we have to acquire funding. Unfortunately, we cannot quantify the 239 effect of the DFIR in this paper.
- In Fig.1 (especially panel a) the MPS seems to detect slightly (but systematically) lower fall
   velocities than the 2DVD in the Greeley data. This is not the case for the Huntsville data (Fig. 3).
   Please give an explanation for this.
- Please see new Table 1 for a quantitative comparison between MPS and 2DVD in the overlap
  region. The agreement between the two instruments is, in our opinion, excellent for both sites
  given the quoted accuracies in fall speed measurement for both instruments.
- 249We have replaced the fit to Gunn-Kinzer data with the 9th order polynomial fit given in Foote and250du Toit (1969) as opposed to using the exponential fit of Atlas et al. (1973) in both figs. 1a and2513a. The latter fit is not as accurate for the small drop end (e.g., it does not pass through the252origin).
- 253

- 254The pressure adjustment to the Gunn-Kinzer fit to account for the altitude (1.4 km MSL) of the255Greeley site is in excellent agreement with the 2DVD measurements. The slight underestimation256of the MPS fall speeds relative to the fit of Foote and du Toit to the data of Gunn-Kinzer in the257range 0.7-1.5 mm (max deviation of 0.5 m/s or 10%) is puzzling given the excellent agreement258for the Huntsville site (sea level). We have re-worded the sentences in Section 2.1 as follows:
- "Panel (a) demonstrates the excellent "visual" agreement between the two
  instruments in the overlap size range (0.7-2 mm) which is quantified in Table 1.
  However, the altitude-adjusted FT fit is slightly higher than the measured values
  as shown in Table 1."
- 4. On lines 208-215, the correlation between E on the one hand, and the mean and standard deviation of the raindrop fall speeds on the other is discussed. I agree that this correlation is there. However, judging from Fig. 4, I think there is also some correlation with the rain rate R (especially the peak at 10 UTC). Please elaborate on the role of the rain rate for these correlations.
- The reviewer is correct in that there is a correlation with rain rate (as shown in the figure below
  for the reviewer's benefit) but this can be misleading in that heavy rain can and does occur
  during calm conditions and the opposite also occurs. So it is difficult to come to a firm conclusion
  unless we look at many cases which is reserved for future work.

259

264

270

275

- Figure 1: The mean fall speed versus rain rate for 30 Nov 2016 Huntsville for 1.3 and 2 mm drops.
- 279 5. On lines 224-225, the observed near-linear decrease of the mean fall speed with turbulence
   280 intensity (or at least its proxy E) is mentioned. How significant is this relation?

282

Not sure if what is meant by "significant" relation. We can do a linear fit and compute correlation coefficient but this won't add much to what is already fairly obvious. We do not wish to quantify this as E is only a proxy for turbulence.

285

290

- 6. On lines 247-254, the results presented by Montero-Martinez et al. (2009) are compared to
  those presented in this paper. What could be the explanation for this difference? Could it be
  something similar to what is discussed in the next paragraph (lines 255-262) about the findings
  of Larsen et al. (2014)? Please elaborate on this.
- We can only speculate why Montero-Martinez et al. found strongly skewed distribution for 0.44 mm drops. The instruments they used, 2D-C and 2D-P, were designed for use on aircraft and not as fixed disdrometers. The airspeed clock is about a factor of 10 higher in their implementation. The calibration method is not discussed in any detail by them whereas the MPS can be calibrated as often as needed with a special device. They did not use any wind shield as far as we can ascertain.
- 298The Larsen et al. study used a 2DVD which has "mis-matched" drop problem not present299in the 2D-C,P probes.

300

305

297

- 3017. On lines 272-281, the relation to the findings of Stout et al. (1995) are discussed. Is there an302empirical relation between E and the rms velocity fluctuations due to turbulence? If so it would303be interesting to see whether the 35% reduction in mean velocity is observed at similar rms304velocity fluctuations-to-terminal fall speed ratios (0.8).
- 306We do not think there is a way to relate our estimate of E based on 3-s wind data to what a 3D-307sonic anemometer would measure in terms of velocity fluctuations at much higher sampling308rate. This for a future project where we would locate a sonic anemometer next to the 2DVD/MPS309inside the DFIR.

310 Technical comments

- 3111. In Figs 2b and 4b, would it be possible to use a second y-axis for R instead of presenting R/10 on312the existing y-axis?
- 313
- 314 Done as suggested.
- 315
- 316 2. On line 275, "greater that" should be "greater than".
- 317 Corrected. Thanks for pointing this out.

- 318 Interactive comment on Atmos. Meas. Tech. Discuss., doi:10.5194/amt-2017-401, 2017.
- Review of AMT-2017-401 By V. Bringi, M. Thurai and D. Baumgardner Manuscript Title Raindrop Fall
- 320 Velocities from an Optical Array Probe and 2D-Video Disdrometer. RC3
- 321
- 322 This manuscript reports on raindrop fall velocity measurements by using two different instruments: a
- 323 MPS (Meteorological Particle Spectrometer) which measures drops in the 0.1-3 mm range, and the
- wildly used 2DVD (two-Dimensional Video Disdrometer), which measures size and fall velocity of drops
- between 0 and 10 mm. The MPS and 2DDVD were used to measure fall velocity of drops in the 0.1-2 and
- larger than 0.7 mm diameter range. The overlapping region 0.7-2 mm diameter was used to cross-
- 327 validate the two measurements. Three different case studies were analyzed in order to relate the
- 328 properties of the drop fall velocity to different precipitation systems (one stratiform, one squall line and
- one super-cell case with low and high turbulence associate for the first two and the third case,
- respectively). The paper is linear and quite easy to read. I have only one major comment that can give a
- 331 contribution, in my opinion, to the generalization of the results. It is reported below together with minor
- 332 comments that, once addressed, will allow the publication of the paper on the Atmospheric
- 333 Measurement Techniques journal.

**334 *The authors appreciate the above general comments and our response is given below in* 335 *italics whereas the text modifications are highlighted.**

336

**337 Major comment.**

- Section 2.2: in the Section 2.1 the authors investigated a stratiform case, while in the Section 2.2 a

squall line and a super-cell case. The squall line case reported generally low rainfall rate and turbulence

340 (comparable to the values registered in the stratiform case). It could be useful, in my opinion add (or

- 341 substitute) a convective event, a sort a middle point between a convective and tornadic case, in order to
- have a general overview of the characteristics of drop fall velocity in a broader range of precipitation
- 343 systems.

In Section 2.2 it is clearly stated that..." About 3 h later several squall-line type storm cells passed over the site from 0700-0900 UTC again with strong winds but considerably lower E values 2-4 m2 s-2 and maximum R of 80 mm h-1. After 1000 UTC the E values were much smaller (< 0.5 m2 s-2) indicating calm conditions. The peak R is also smaller at 30 mm h-1 at 1000 UTC." Hence, Fig. 4b already depicts high and moderate-low rain rate conditions. Perhaps the reviewer overlooked the rain rate scale (where we plotted R/10) which has now been changed, the values now on the right Y-axis without any scaling.

351

352

| 354                      |                                                                                                                                                                                                                                                                                                                                                                          |
|--------------------------|--------------------------------------------------------------------------------------------------------------------------------------------------------------------------------------------------------------------------------------------------------------------------------------------------------------------------------------------------------------------------|
| 355                      |                                                                                                                                                                                                                                                                                                                                                                          |
| 356                      |                                                                                                                                                                                                                                                                                                                                                                          |
| 357                      | Minor comments.                                                                                                                                                                                                                                                                                                                                                          |
| 358                      |                                                                                                                                                                                                                                                                                                                                                                          |
| 359
360               | - Line 141: what does it mean that the finite bin width causes a spread of 0.5 m/s? Can the authors explain better? The same is reported in other parts of the text.                                                                                                                                                                                                     |
| 361
362
363        | The bin width for the histograms is $\pm$ 0.1 mm about the center value of 0.5 mm, for example in Fig. 1b all fall speeds that fall in the range 0.4 to 0.6 mm are included in the histogram. The spread of 0.5 m/s is just V(0.6 mm)-V(0.4 mm). In Section 2.1 we have clarified as:                                                                                    |
| 364
365
366        | "The finite bin width used (0.9-1.1 mm) causes a corresponding fall speed "spread" of around 0.5 m s -1 which is clearly a significant contributor to the measured coefficient of variation."                                                                                                                                                                 |
| 367                      | Please see new Table 1 which quantifies the finite bin width spread in Column 1.                                                                                                                                                                                                                                                                                         |
| 368
369               | - Lines 211-215: what is the explanation that the authors give to the decrease of fall speed during the most intense wind and rainfall rate? Does it can be related to the presence of ascending flow?                                                                                                                                                                   |
| 370
371
372
373 | In Section 3 we refer to Stout et al. (1995) who have simulated the effects of turbulence to cause a decrease in fall speed relative to still air conditions. The effect is due to increase in the non-linear drag due to both vertical and horizontal gusts. We defer to the explanation in Stout et al. (1995) but we introduce this reference earlier in Section 2.2. |
| 374
375               | - Lines 224-225: similar to the previous comment. How do they justify the decrease of fall speed when E
(turbulence) increases?                                                                                                                                                                                                                                       |
| 376                      | Please refer to our response above.                                                                                                                                                                                                                                                                                                                                      |
| 377                      |                                                                                                                                                                                                                                                                                                                                                                          |
| 378
379               | - Panel (b) of Figures 2 and 4: the rain rate should be reported on the right y-axis avoiding the necessity to show its values scaled on a factor ten.                                                                                                                                                                                                                   |
| 380                      | Done as requested.                                                                                                                                                                                                                                                                                                                                                       |
| 381                      |                                                                                                                                                                                                                                                                                                                                                                          |
| 382                      | - Figure 2a: the y-axis limit should not exceed 10 m/s to improve the detail of the plot.                                                                                                                                                                                                                                                                                |
| 383                      | Done as requested.                                                                                                                                                                                                                                                                                                                                                       |

385

Interactive comment on "Raindrop Fall Velocities from an Optical Array Probe and 2D-video
 Disdrometer" by Viswanathan Bringi et al.

388 Anonymous Referee #4 Received and published: 18 December 2017: RC4

389 GERNERAL COMMENTS This is an excellent paper, addressing the issue of in-situ measurements of rain

drops over their full size range, with focus on the drops' fall velocities. Especially the characteristics of

- 391 very small drops still are not investigated thoroughly up to here. The investigation of the relationship
- between wind turbulence and rain drops' fall velocity is convincing and in the future will allow new
- refinements in the related remote sensing algorithms. The study is presented in a clear and concise way.
- 394 The reviewer likes to point out, that such work does not only address academic interests, but points to a
- number of applications, like remote sensing of the atmosphere, as the authors shortly mention in their
- 396 introduction. Other applications include knowledge of channel characteristics for satellite
- 397 communications at EHF frequency bands. Thus the study is very relevant for these fields of science and
- applications. The work is based on the combined analyses of data from 2 instruments at same site, the
- 399 Meteorological Particle Spectrometer (MPS) and the 2D-video disdrometer (2DVD). Field measurement
- 400 of rain drops still may be considered as challenge, with quite a few solutions being around, but none of
- 401 the instruments may claim to be perfect. The presented results do rise a few relevant questions, which
- 402 are given in below specific comments. Summarizing it is said, that this paper gives an excellent
- 403 contribution, bringing new aspects for the relevant fields of science and applications.

**We appreciate the very positive comments by the reviewer. Our response is in italics below and modifications to the text are highlighted.**

406 SPECIFIC COMMENTS

\*\*\* Line 98 / 99: "All possible corrections have been applied, including the removal of artifacts due to
splashing,..." Careful preprocessing, verification and validation of data is of utmost importance for the
present study. The results presented require correctness of challenging measurement processes. Some
of the result given actually leave a few questions open, as addressed in below comment to Fig 1a and
Fig3a. To be sufficiently self-contained, thus the paper might shortly describe the mentioned correction
algorithms.

- We have replaced the sentence: "All possible corrections have been applied, including the removal of
  artifacts due to splashing, and oversizing that results from out-of-focus droplets (Korolev 2007)." With
- 415 *"There are a number of potential artifacts that arise when making measurements with optical array*
- 416 probes (Baumgardner et al., 2017): droplet breakup on the probe tips that form satellite droplets,
- 417 multiple droplets imaged simultaneously, and out-of-focus drops whose images are usually larger than
- 418 the actual drop (Korolev, 2007). The measured images have been analyzed to remove satellite droplets
- 419 whose interarrival times are usually too short to be natural drops, multiple drops are detected by shape

- 420 analysis and removed, and out-of-focus drops are detected and size corrected using the technique
  421 described by Korolev (2007)."
- 422 \*\*\* Line 157: "The histogram from MPS for the 0.5 mm sizes shows positive skewness" Can the authors423 give an explanation / discussion / assumption?

424 The histogram in Fig. 1b for 0.5 mm size is skewed towards higher values...the tail extends

from 2.6 to 4 m/s. Given that the mean is 1.8 m/s and  $\sigma$ =0.65 m/s, there is a finite occurrence of

426 super-terminal fall speeds that exceed mean+1σ or 2.45 m/s. Such super-terminal speeds have

- 427 been noted by Montero-Martinez et al using 2DC,P probes for similar sized drops in fact with
- 428 much longer tail or skewness than our observations.
- \*\*\* Fig. 1a and Fig. 3a: Mean fall velocities from the MPS are read from these figures approximately as:
  Fig. 1a: D = 0,7 mm, v = 2.55 m/s
- 431 Fig. 3a: D = 0.7 mm, v = 3.05 m/s
- 432 That represents an exceedance by more than 16 % (Fig. 3a over Fig. 1a), in spite of the lower pressure in
- Greely (Fig 1a) leading to the expectation of faster drops than in Huntsville (Fig 3a). The authors please
  could discuss this.
- 435 We refer to new Table 1 from which we now have:
- 436 Greeley site D=0.7 mm mean MPS=2.6 m/s
- 437 Huntsville D=0.7 mm mean MPS=2.6 m/s
- 438 We have also added new histograms for 0.7 and 1.5 mm from both sites (new Fig. 1c and 3c).
- 439 The 0.7 mm histogram from Greeley is slightly skewed (Fig. 1c) whereas the Hunsville
- 440 histogram is more symmetric. For small drops (< 1 mm) the pressure adjustment is not
- 441 significant (see Fig. 1a) and this is reflected by Table 1 data.
- 442
- 443 \*\*\* Fig. 3a: Mean fall velocities from the MPS are read from this figure approximately as:
- 444 D = 1.5 mm, v = 5.28 m/s
- 445 D = 1.6 mm, v = 6.31 m/s
- 446 D = 1.7 mm, v = 6.5 m/s
- 447 D = 1.8 mm, v = 5.93 m/s
- 448 These values differ significantly from the fit to Gunn-Kinzer, further from the expected monotonic
- 449 behaviour. The authors please could discuss this.

450 We have added a new Table 1 which quantifies the measurement comparison between MPS and 2DVD in

451 the overlap region for the 2 sites. The agreement in the mean fall speed is excellent for both sites. The

452 mean values quoted above by the reviewer are not accurate but we do appreciate the effort in volved to

453 read values of a graph. In particular at D=1.7 mm, from Table 1 the mean is 6±0.3 m/s (and not 5.93 m/s

454 as quoted above). At D=1.8 mm, interpolation gives 6.25 m/s which is monotonic behavior. Table 1 also

455 gives the expected values using the Foote and du Toit (1969) 9th order polynomial fit (FT) to Gunn-Kinzer

456 which is more accurate than the exponential fit of Atlas et al especially for the smaller drops. We list

457 below the FT fit value and mean MPS from Table 1 for Huntsville site.

| 458 | 0.7 mm | FT fit=2.9 m/s | MPS=2.6 | 2DVD=2.5           |                  |
|-----|--------|----------------|---------|--------------------|------------------|
| 459 | 0.9    | 3.65           | 3.4     | 3.3                |                  |
| 460 | 1.1    | 4.3            | 4.2     | 4.1                |                  |
| 461 | 1.3    | 4.9            | 4.9     | 4.9                |                  |
| 462 | 1.5    | 5.45           | 5.4     | 5.4 (see new histo | gram in Fig. 3c) |
| 463 | 1.7    | 5.9            | 6.0     | 5.8                |                  |
| 464 | 1.9    | 6.3            | 6.5     | 6.3                |                  |

465 For 0.7 and 0.9 mm sizes the MPS and 2DVD mean values are systematically lower than the FT fit

by 7-10%. For comparison, Yu et al. (2016; new reference added in revised text) found that their high
speed camera measurement of fall speeds in the interior stair-case of a building also were systematically
lower by 5% (for sizes in the range 0.4 to 1.6 mm) relative to the same FT fit as used here (their fig. 7).

469 TECHNICAL COMMENTS:

470 \*\*\* Line 230: "(σf) versus E is shown in panels 6 (b,d)." It probabaly should read as "(σf) versus E is
471 shown in panels (b,d) of Fig 5."

472 Done...thank you for pointing this out.

473 Interactive comment on Atmos. Meas. Tech. Discuss., doi:10.5194/amt-2017-401, 2017.

474

- 476
- 477
- 478

- Raindrop Fall Velocities from an Optical Array Probe and 2D-Video
  Disdrometer
- 482 Viswanathan Bringi1, Merhala Thurai1 and Darrel Baumgardner2

1 Department of Electrical and Computer Engineering, Colorado State University, Fort
 Collins, Colorado, USA

- 485 2 Droplet Measurements Technologies, Longmont, Colorado, USA
- 486 Correspondence to: V.N. Bringi
- 487 Email: bringi@colostate.edu

488

**490 Abstract**

We report on fall speed measurements of rain drops in light-to-heavy rain events from 491 two climatically different regimes (Greeley, Colorado, and Huntsville, Alabama) using 492 the high resolution (50 µm) Meteorological Particle Spectrometer (MPS) and a 3rd 493 generation (170 µm resolution) 2D-video disdrometer (2DVD). To mitigate wind-effects, 494 especially for the small drops, both instruments were installed within a 2/3-scale Double 495 Fence Intercomparison Reference (DFIR) enclosure. Two cases involved light-to-496 moderate wind speeds/gusts while the third case was a tornadic supercell and several 497 squall-lines that passed over the site with high wind speeds/gusts. As a proxy for 498 turbulent intensity, maximum wind speeds from 10-m height at the instrumented site 499 recorded every 3 s were differenced with the 5-min average wind speeds and then 500 squared. The fall speeds versus size from 0.1-2 mm and >0.7 mm were derived from 501 the MPS and the 2DVD, respectively. Consistency of fall speeds from the two 502 instruments in the overlap region (0.7-2 mm) gave confidence in the data quality and 503 processing methodologies. Our results indicate that under low turbulence, the mean fall 504 505 speeds agree well with fits to the terminal velocity measured in the laboratory by Gunn and Kinzer from 100 µm up to precipitation sizes. The histograms of fall speeds for 0.5, 506 507 0.7, 1 and 1.5 mm sizes were examined in detail under the same conditions. The histogram shapes for the 1 and 1.5 mm sizes were symmetric and in good agreement 508 509 between the two instruments with no evidence of skewness or of sub- or super-terminal fall speeds. The histograms of the smaller 0.5 and 0.7 mm drops from MPS while 510 generally symmetric showed that occasional occurrences of sub- and super-terminal fall 511 speeds could not be ruled out. In the supercell case, the very strong gusts and inferred 512 high turbulence intensity caused a significant broadening of the fall speed distributions 513 with negative skewness (for drops of 1.3, 2 and 3 mm). The mean fall speeds were also 514 found to decrease nearly linearly with increasing turbulent intensity attaining values 515 about 25-30% less than the terminal velocity of Gunn-Kinzer, i.e. sub-terminal fall 516 517 speeds.

518

**520 1 Introduction**

Knowledge of the terminal fall speed of raindrops as a function of size is important in 521 522 modelling collisional break-up and coalescence processes (e.g., List et al., 1987), in the 523 radar-based estimation of rain rate, in retrieval of drop size distribution using Doppler spectra at vertical incidence (e.g., Sekhon and Srivastava, 1971) and in soil erosion 524 studies (e.g., Rosewell 1986). In these and other applications it is generally accepted 525 that there is a unique fall speed ascribed to drops of a given mass or diameter and that 526 527 it equals the terminal speed with adjustment for pressure (e.g., Beard 1976). The terminal velocity measurements of Gunn and Kinzer, 1949) under calm laboratory 528 conditions, and fits to their data (e.g., Atlas et al., 1973; Foote and du Toit, 1969; Beard 529 and Pruppacher, 1969) are still considered the standard against which measurements 530 using more modern optical instruments in natural rain are compared (Löffler-Mang and 531 Joss, 2000; Barthazy et al., 2004; Schönhuber et al., 2008; Testik and Rahman, 2016; 532 Yu et al., 2016). More recently, the broadening and skewness of the fall speed 533 distributions of a given size (3 mm) in one intense rain event were attributed to mixed-534 535 mode amplitude oscillations (Thurai et al., 2013). Super- and sub-terminal fall speeds in intense rain shafts have been detected and attributed, respectively, to drop breakup 536 fragments (sizes < 0.5 mm), and high wind/gusts (sizes 1-2 mm) (Montero-Martinez et 537 al., 2009; Larsen et al., 2014; Montero-Martinez and Garcia-Garcia, 2016). Thus, there 538 is some evidence that rain drops may not fall at their terminal velocity except under 539 calm conditions and that the concept of a fall speed distribution for a drop of given mass 540 (or, diameter) might need to be considered which is the topic of this paper. The 541 implications are rather profound especially for numerical modeling of collision-542 coalescence and breakup processes which are important for shaping the drop size 543 distribution. 544

The fall speeds and concentration of small drops (< 1 mm) in natural rain are difficult to 545 measure accurately given the poor resolution (>170 µm) of most optical disdrometers 546 and/or sensitivity issues. While cloud imaging probes (with high resolution 25-50 µm) 547 on aircraft have been used for many years they generally cannot measure the fall 548 549 speeds. A relatively new instrument, the Meteorological Particle Spectrometer (MPS) is a droplet imaging probe that was built by Droplet Measurements Technologies (DMT, 550 Inc.) under contract from the US Weather Service specifically designed for drizzle as 551 small as 50 µm and rain drops up to 3 mm. This instrument in conjunction with a lower 552 553 resolution 2D-Video Disdrometer (Schoenhuber et al., 2008) is used in this paper to 554 measure fall speed distributions in natural rain.

555 This paper briefly describes the instruments used, presents fall speed measurements 556 from two sites under relatively low wind conditions, and one case from an unusual tornadic supercell with high winds and gusts and ends with a brief discussion and summary of the results.

559

**560 2 Instrumentation and Measurements**

The principal instruments used in this study are the MPS and 3rd generation 2D-video 561 disdrometer (2DVD), both located within a 2/3-scale Double Fence Intercomparison 562 Reference (DFIR: Rasmussen et al., 2012) wind shield. As reported in (Notaros et al., 563 2016), the 2/3-scale DFIR was effective in reducing the ambient wind speeds by nearly 564 a factor of 2-3 based on data from outside and inside the fence. The flow field in and 565 566 around the DFIR has been simulated by (Theriault et al., 2015) assuming steady 567 ambient winds. They found that depending on the wind direction relative to the octagonal fence, weak vertical motions could be generated above the sensor areas. For 568 5 m/s speeds, the motions could range between -0.4 (down draft) to 0.2 m/s (up draft). 569

570 The instrument set-up was the same for the two sites (Greeley, Colorado and 571 Huntsville, Alabama). Huntsville has a very different climate from Greeley, and its 572 altitude is 212 m MSL as compared with 1.4 km MSL for Greeley. According to the 573 Köppen–Trewartha climate classification system (*Trewartha and Horn,* 1980), this labels 574 Greeley as a semiarid-type climate, whereas Huntsville is a humid subtropical-type 575 climate (*Belda et al.,* 2014).

The MPS is an optical array probe (OAP) that uses the technique introduced by 576 Knollenberg (1970, 1976, 1980) and measures drop diameter in the range from 0.05-3.1 577 mm. A 64 element photo-diode array is illuminated with a 660 nm collimated laser 578 579 beam. Droplets passing through the laser cast a shadow on the array and the decrease 580 in light intensity on the diodes is monitored with the signal processing electronics. A two dimensional image is captured by recording the light level of each diode during the 581 period that the array is shadowed. The fall velocity is derived using two methods. One 582 uses the same approach as described by (Montero-Martinez et al., 2009) where the fall 583 velocity is calculated from the product of the true air speed clock and ratio of the image 584 height -to-width. Note that "width" is the horizontal dimension parallel to the array and 585 "height" is along the vertical. The second method computes the fall velocity from the 586 maximum horizontal dimension (spherical drop shape assumption) divided by the 587 amount of time that the image is on the array, a time measured with a 2 MHz clock. In 588 order to be comparable to the results of (Montero-Martinez et al., 2009), their approach 589 590 is implemented here for sizes > 250  $\mu$ m. The fall velocity of smaller, slower moving 591 droplets, is measured using the second technique.

The limitations and uncertainties associated with OAP measurements have been well 592 documented (Korolev et al., 1991; 1998; Baumgardner et al., 2017). There are a 593 number of potential artifacts that arise when making measurements with optical array 594 probes (Baumgardner et al., 2017): droplet breakup on the probe tips that form satellite 595 596 droplets, multiple droplets imaged simultaneously, and out-of-focus drops whose images are usually larger than the actual drop (Korolev, 2007). The measured images 597 have been analyzed to remove satellite droplets whose interarrival times are usually too 598 short to be natural drops, multiple drops are detected by shape analysis and removed, 599 600 and out-of-focus drops are detected and size corrected using the technique described by (Korolev 2007). The sizing and fall speed errors primarily depend on the digitization 601 error ( $\pm$  25 µm). The fall speed accuracy according to the manufacturer (DMT) is <10% 602 for 0.25 mm and <1% for sizes greater than 1 mm, limited primarily by the accuracy in 603 604 droplet sizing.

The 3rd generation 2DVD is described in detail by (Schoenhuber et al., 2007; 2008) and 605 its accuracy of size and fall speed measurement has been well documented (e.g., 606 Thurai et al., 2007; 2009; Huang et al., 2008; Bernauer et al., 2015). Considering the 607 horizontal pixel resolution of 170 µm and other factors (such as "mis-matched" drops), 608 the effective sizing range is D> 0.7 mm. To clarify the "mis-matched" drop problem: it is 609 very difficult to match a drop detected in the top light-beam plane of the 2DVD to the 610 corresponding drop in the bottom plane for tiny drops resulting in erroneous fall speeds. 611 The fall velocity accuracy is determined primarily by the accuracy of calibrating the 612 distance between the two orthogonal light "sheets" or planes and is < 5% for fall velocity 613 In our application, we utilize the MPS for measurement of small drops 614 <10 m s-1. with D < 1.2 mm. The measurements from the MPS are compared with those from the 615 2DVD in the overlap region of  $D \approx 0.7-2.0$  mm to ensure consistency of 616 observations. The only fall velocity threshold used for the 2DVD is the lower limit set at 617 0.5 m s-1 in accordance with the manufacturer guidelines for rain measurements. 618

619 2.1 Fall Speeds from Greeley, Colorado

We first consider a long duration (around 20 h) rain episode on 17 April 2015 which consisted of a wide variety of rain types/rates (mostly light stratiform < 8 mm h-1) as described in Table 2 of (*Thurai et al.*, 2017). Two wind sensors at a height of 1 m were available to measure the winds outside and inside the DFIR. Average wind speeds were, respectively, < 1.5 m s-1 inside the DFIR and < 4 m s-1 outside with light gusts. These wind sensors were specific to the winter experiment described in (*Notaros et al.*, 2016) and were unavailable for the rain measurement campaign after May 2015.

Figure 1(a) shows the fall speeds versus D from the 2DVD (shown as contoured frequency of occurrence), along with mean and  $\pm 1\sigma$  standard deviation from the MPS. Also shown is the (*Foote and du Toit 1969*) (henceforth FT fit) to the terminal fall speed

measurements of (Gunn and Kinzer, 1949) at sea level and after applying altitude 630 corrections (*Beard*, 1976) for the elevation of 1.4 km MSL for Greeley. Panels (b,c) 631 shows the histogram of fall speeds for diameter intervals (0.5±0.1) and (1±0.1 mm), and 632 (0.7±0.1) and (1.5±0.1 mm), respectively. Panel (a) demonstrates the excellent "visual" 633 634 agreement between the two instruments in the overlap size range (0.7-2 mm) which is quantified in Table 1. However, the altitude-adjusted FT fit is slightly higher than the 635 measured values as shown in Table 1. Notable in Fig. 1a is the remarkable agreement 636 in mean fall speeds between the FT fit and the MPS for D< 0.5 mm down to near the 637 lower limit of the instrument (0.1 mm). Few measurements have been reported of fall 638 speeds in this size range. 639

Table 1: Expected fall velocities for various diameter intervals (bin width of 0.2 mm) from (*Foote and du Toit*, 1969) with altitude adjustment, and the measured mean fall velocities with  $\pm 1\sigma$  (standard deviation)

| D range (mm) | Expected (m s -1 ) | MPS (m s -1)       | 2DVD (m s -1 )     |
|--------------|-------------------------------|-------------------------------|-------------------------------|
| (Greeley)    | at 1.4 km                     | Mean $\pm 1\sigma$            | Mean $\pm 1\sigma$            |
| 0.6 to 0.8   | 2.6 to 3.5                    | $\textbf{2.6}\pm\textbf{0.6}$ | $2.5\pm0.8$                   |
| 0.8 to 1.0   | 3.5 to 4.3                    | $\textbf{3.4}\pm\textbf{0.6}$ | $3.3\pm0.9$                   |
| 1.0 to 1.2   | 4.3 to 4.9                    | $\textbf{4.2}\pm\textbf{0.6}$ | $4.1\pm0.9$                   |
| 1.2 to 1.4   | 4.9 to 5.5                    | $4.9\pm0.5$                   | $5.0\pm0.8$                   |
| 1.4 to 1.6   | 5.5 to 6.1                    | $5.6\pm0.5$                   | 5.7 ± 0.7                     |
| 1.6 to 1.8   | 6.1 to 6.6                    | $\textbf{6.1}\pm\textbf{0.4}$ | $6.2\pm0.7$                   |
| 1.8 to 2.0   | 6.6 to 7.0                    | $\textbf{6.7}\pm\textbf{0.4}$ | $6.6\pm0.8$                   |
|              |                               |                               |                               |
| D range (mm) | Expected (m/s)                | MPS (m/s)                     | 2DVD (m/s)                    |
| (Huntsville) | at 0 km                       | Mean $\pm$ Std_dev            | Mean ± Std. dev               |
| 0.6 to 0.8   | 2.5 to 3.3                    | $\textbf{2.6}\pm\textbf{0.6}$ | $2.5\pm0.7$                   |
| 0.8 to 1.0   | 3.3 to 4.0                    | $\textbf{3.4}\pm\textbf{0.5}$ | $\textbf{3.3}\pm\textbf{0.7}$ |
| 1.0 to 1.2   | 4.0 to 4.6                    | $\textbf{4.2}\pm\textbf{0.6}$ | $4.1\pm0.8$                   |
| 1.2 to 1.4   | 4.6 to 5.2                    | $\textbf{4.9}\pm\textbf{0.4}$ | $4.9\pm0.7$                   |
| 1.4 to 1.6   | 5.2 to 5.7                    | $5.4\pm0.4$                   | $5.4\pm0.6$                   |
| 1.6 to 1.8   | 5.7 to 6.1                    | $6.0\pm0.3$                   | $5.8\pm0.6$                   |
| 1.8 to 2.0   | 6.1 to 6.5                    | $6.5\pm0.4$                   | $6.3\pm0.5$                   |

643

The histograms in Fig. 1(b,e) show good agreement between 2DVD and MPS for 1 mm and 1.5 mm drop sizes, respectively, with respect to the mode, symmetry, spectral width and lack of skewness in the distributions. For the 1 mm size histogram, the mean is 3.8 m s-1 while the spectral width or standard deviation from MPS data is 0.6 m s-1. The corresponding coefficient of variation (ratio of standard deviation to mean) is 15.7%. The finite bin width used (0.9-1.1 mm) causes a corresponding fall speed "spread" of around 0.6 m s-1 which is clearly a significant contributor to the measured coefficient of

variation. Similar comments apply to the fall speed histogram for the 1.5 mm size shown 651 in Fig. 1c. The definition of sub- or super-terminal fall speeds by (Montero-Martinez et 652 al., 2009) is based on fall speeds that are, respectively, less than 0.7 times the mean 653 value or greater than 1.3 times the mean value (i.e., exceeding 30% threshold on either 654 655 side of the mean terminal fall speed). From examining the 1 mm size fall speed histogram there is negligible evidence of occurrences with fall speeds < 2.66 m s-1 656 (sub) or > 4.94 m s-1 (super). Similar comment also applies for the 1.5 mm size based 657 on the corresponding histogram. 658

---

## Referee Comment (RC5) · Anonymous Referee #5 · 9 Jan 2018

This manuscript is a summary of observations of the fall speeds of small raindrops near the ground, in a field setting. It is comforting to see that the observations are generally consistent with the familiar Gunn and Kinzer laboratory measurements in conditions not strongly affected by wind and turbulence. In the latter situation the measured fall speeds tended to be less, and therein lays a puzzle that warrants more discussion in the manuscript. Certainly turbulence in the low atmosphere could increase the spread of the drop fall speeds, but it does not produce a significant vertical mass flux. If the turbulence were isotropic (which may not be the case here) Stout et al. did find indications of reduced fall speeds. However, it's not clear why there would not be a similar neutral effect on the overall raindrop flux. If the drops are falling at normal terminal speeds in the free atmosphere and at reduced speeds near the surface there would

have to be an accumulation of rain at some level above the disdrometer. Of course there might be intermittent episodes of "super-" and "sub-terminal" fall speeds but the duration of the latter in the observations is a substantial fraction of an hour. I do not have an explanation but invite the authors to offer one, or at least discuss the subject. My other comments are relatively minor: There are instances of singular-plural subject-verb disagreement in the manuscript. L15ff: "micron" is not an SI unit (and compare with L57). L41: The authors might add the very useful fall speed relationship from Uplinger (Uplinger WG. 1989. A new formula for raindrop terminal velocity. Preprints, 20th Conference Radar Meteorology, 389–391). L81-103 (maybe 105-115 as well): What about possible edge effects on the measurements? Under the conditions listed in L199-207, a 1 mm drop at terminal fall speed would approach the instrument at an angle of <22 deg from horizontal. Viewing the measurement plane from that angle, there's a lot more edge than when the approach is vertical. L89: How does the "true air speed clock" work in this situation? L162: The expression in parentheses is not what is really meant. L230: "...panel 5 (b,d)." L251-252: If these (1 and 5) are fall speeds, include the units. Fig. 3a: Any clue what caused the MPS hiccup after 1.5 mm size?

---

## Author Comment (AC2) · 31 Jan 2018

**Response to Reviewer # 5**

***We appreciate the reviewer's comments and our response is given in blue bold italics. The modifications to revised manuscript based on earlier 4 reviews (major revision) are attached and highlighted. The modifications of the manuscript based on this last review are very minor but included also.***

*This manuscript is a summary of observations of the fall speeds of small raindrops near the ground, in a field setting. It is comforting to see that the observations are generally consistent with the familiar Gunn and Kinzer laboratory measurements in conditions not strongly affected by wind and turbulence.*

In the latter situation the measured fall speeds tended to be less, and therein lays a puzzle that warrants more discussion in the manuscript. Certainly turbulence in the low atmosphere could increase the spread of the drop fall speeds, but it does not produce a significant vertical mass flux. If the turbulence were isotropic (which may not be the case here) Stout et al. did find indications of reduced fall speeds. However, it's not clear why there would not be a similar neutral effect on the overall raindrop flux. If the drops are falling at normal terminal speeds in the free atmosphere and at reduced speeds near the surface there would have to be an accumulation of rain at some level above the disdrometer. Of course there might be intermittent episodes of "super-" and "sub-terminal" fall speeds but the duration of the latter in the observations is a substantial fraction of an hour. I do not have an explanation but invite the authors to offer one, or at least discuss the subject.

***Response to General Comments: What the reviewer is assuming as best as we can ascertain is related to the sedimentation of raindrops i.e., time rate of change of mass equals negative of the height derivative of mVt. This neglects other important processes such as evaporation and source/sink terms for coalescence-breakup. Plus the fall speeds increase at lower pressure aloft. We do not feel we can offer any more explanation and hence no modifications to the manuscript have been made.***

*My other comments are relatively minor:*

There are instances of singular-plural subject-verb disagreement in the manuscript.

***The manuscript has been revised substantially in response to Reviews 1-4 which is now attached to this response***.

L15ff: "micron" is not an SI unit (and compare with L57).

***We have changed 'microns' to 'μm' throughout the text.***

L41: The authors might add the very useful fall speed relationship from Uplinger (Uplinger WG. 1989. A new formula for raindrop terminal velocity. Preprints, 20th Conference Radar Meteorology, 389–391).

*In our revised manuscript we have used the fit of Foote and du Toit (1969) to replace the fit of Atlas et al. (1973). The figure below compares the Foote and du Toit fit to Uplinget (1977). Since the agreement is excellent we feel no need to use the latter fit and hence no modifications have been made to the manuscript.*

[Figure]

L81-103 (maybe 105-115 as well): What about possible edge effects on the measurements?

Under the conditions listed in L199-207, a 1 mm drop at terminal fall speed would approach the instrument at an angle of <22 deg from horizontal. Viewing the measurement plane from that angle, there's a lot more edge than when the approach is vertical.

*The 2DVD and MPS instruments were collocated inside a DFIR wind shield so the drop trajectories are different as opposed to un-shielded case. Note also that the DFIR reduces the wind speed at the center of the fence by nearly a factor of three relative to the environment wind speed. The edge effect under strong horizontal winds can be easily noted by partial filling of the 2DVD's 10X10 cm sensor area which is not the case as shown in figure below, i.e., the 2 mm drops uniformly fill the sensor area during the high wind period of the squall lines (0600-1000 UTC) in Fig. 4.*

[Figure]

30Nov2016, 06–10 hr, 1.9–2.1 mm drops

*Each point is the location of drop in the 10X10 cm sensor area of the 2DVD. All drops in the range 1.9-2.1 mm during the time segment 0600-1000 UTC on 30 Nov 2016 are shown.*

L89: How does the "true air speed clock" work in this situation?

*We understand the question posed by the reviewer although the effect that the reviewer is referring to will depend on the trajectory of the drop with respect to the diode array. We use the fin (wind vane) to minimize the angle issue, i.e. if the array is completely perpendicular to the trajectory of the drop, the horizontal motion of the drop will have no impact. If, however, in the worse case the drop trajectory is parallel to the array, there can be a potential for oversizing but the error will still be quite small since the amount of time that it takes to capture the image slice is very small compared to how far across the array a droplet travels horizontally during that time. In other words, the captured image appears as if it is going through at an angle, but the maximum width will still be the same as the drop.*

*We note that in our paper the MPS was configured without a wind vane as it was sited inside the DFIR wind shield. The diode array was oriented perpendicular to the environmental mean wind direction for both Greeley and Huntsville site.*

[Figure]

*Sample of drop images recorded by the MPS. Schematic of drop whose trajectory is parallel to the diode array.*

L162: The expression in parentheses is not what is really meant.

*We have deleted the expression in parenthesis as it is not necessary. Thank you for pointing this out.*

L230: " ...panel 5 (b,d)."

*Corrected. Thank you.*

L251-252: If these (1 and 5) are fall speeds, include the units. Fig. 3a: Any clue what caused the MPS hiccup after 1.5 mm size?

*Units inserted as suggested. The revised figure does not have the "hiccup".*

**Raindrop Fall Velocities from an Optical Array Probe and 2D-Video Disdrometer**

Viswanathan Bringi[1], Merhala Thurai[1] and Darrel Baumgardner[2]

[1] *Department of Electrical and Computer Engineering, Colorado State University, Fort Collins, Colorado, USA*

[2] *Droplet Measurements Technologies, Longmont, Colorado, USA*

Correspondence to: V.N. Bringi

Email: bringi@colostate.edu

**Abstract**

We report on fall speed measurements of rain drops in light-to-heavy rain events from two climatically different regimes (Greeley, Colorado, and Huntsville, Alabama) using the high resolution (50 μm) Meteorological Particle Spectrometer (MPS) and a 3$^{rd}$ generation (170 μm resolution) 2D-video disdrometer (2DVD). To mitigate wind-effects, especially for the small drops, both instruments were installed within a 2/3-scale Double Fence Intercomparison Reference (DFIR) enclosure. Two cases involved light-to-moderate wind speeds/gusts while the third case was a tornadic supercell and several squall-lines that passed over the site with high wind speeds/gusts. As a proxy for turbulent intensity, maximum wind speeds from 10-m height at the instrumented site recorded every 3 s were differenced with the 5-min average wind speeds and then squared. The fall speeds versus size from 0.1-2 mm and >0.7 mm were derived from the MPS and the 2DVD, respectively. Consistency of fall speeds from the two instruments in the overlap region (0.7-2 mm) gave confidence in the data quality and processing methodologies. Our results indicate that under low turbulence, the mean fall speeds agree well with fits to the terminal velocity measured in the laboratory by Gunn and Kinzer from 100 μm up to precipitation sizes. The histograms of fall speeds for 0.5, 0.7, 1 and 1.5 mm sizes were examined in detail under the same conditions. The histogram shapes for the 1 and 1.5 mm sizes were symmetric and in good agreement between the two instruments with no evidence of skewness or of sub- or super-terminal fall speeds. The histograms of the smaller 0.5 and 0.7 mm drops from MPS while generally symmetric showed that occasional occurrences of sub- and super-terminal fall speeds could not be ruled out. In the supercell case, the very strong gusts and inferred high turbulence intensity caused a significant broadening of the fall speed distributions with negative skewness (for drops of 1.3, 2 and 3 mm). The mean fall speeds were also found to decrease nearly linearly with increasing turbulent intensity attaining values about 25-30% less than the terminal velocity of Gunn-Kinzer, i.e. sub-terminal fall speeds.

**1 Introduction**

Knowledge of the terminal fall speed of raindrops as a function of size is important in modelling collisional break-up and coalescence processes (e.g., *List et al.,* 1987), in the radar-based estimation of rain rate, in retrieval of drop size distribution using Doppler spectra at vertical incidence (e.g., *Sekhon and Srivastava,* 1971) and in soil erosion studies (e.g., *Rosewell* 1986). In these and other applications it is generally accepted that there is a unique fall speed ascribed to drops of a given mass or diameter and that it equals the terminal speed with adjustment for pressure (e.g., *Beard* 1976). The terminal velocity measurements of *Gunn and Kinzer,* 1949) under calm laboratory conditions, and fits to their data (e.g., *Atlas et al.,* 1973; *Foote and du Toit,* 1969; *Beard and Pruppacher,* 1969) are still considered the standard against which measurements using more modern optical instruments in natural rain are compared (*Löffler-Mang and Joss,* 2000; *Barthazy et al.,* 2004; *Schönhuber et al*., 2008; *Testik and Rahman,* 2016; *Yu et al.,* 2016). More recently, the broadening and skewness of the fall speed distributions of a given size (3 mm) in one intense rain event were attributed to mixed-mode amplitude oscillations (*Thurai et al*., 2013). Super- and sub-terminal fall speeds in intense rain shafts have been detected and attributed, respectively, to drop breakup fragments (sizes < 0.5 mm), and high wind/gusts (sizes 1-2 mm) (*Montero-Martinez et al*., 2009; *Larsen et al.,* 2014; *Montero-Martinez and Garcia-Garcia*, 2016). Thus, there is some evidence that rain drops may not fall at their terminal velocity except under calm conditions and that the concept of a fall speed distribution for a drop of given mass (or, diameter) might need to be considered which is the topic of this paper. The implications are rather profound especially for numerical modeling of collision-coalescence and breakup processes which are important for shaping the drop size distribution.

[revised manuscript text omitted]

measurements of (*Gunn and Kinzer*, 1949) at sea level and after applying altitude corrections (*Beard,* 1976) for the elevation of 1.4 km MSL for Greeley. Panels (b,c) shows the histogram of fall speeds for diameter intervals (0.5±0.1) and (1±0.1 mm), and (0.7±0.1) and (1.5±0.1 mm), respectively. Panel (a) demonstrates the excellent "visual" agreement between the two instruments in the overlap size range (0.7-2 mm) which is quantified in Table 1. However, the altitude-adjusted FT fit is slightly higher than the measured values as shown in Table 1. Notable in Fig. 1a is the remarkable agreement in mean fall speeds between the FT fit and the MPS for D< 0.5 mm down to near the lower limit of the instrument (0.1 mm). Few measurements have been reported of fall speeds in this size range.

Table 1: Expected fall velocities for various diameter intervals (bin width of 0.2 mm) from (*Foote and du Toit*, 1969) with altitude adjustment, and the measured mean fall velocities with ±1σ (standard deviation)

| D range (mm) (Greeley) | Expected (m s⁻¹) at 1.4 km | MPS (m s⁻¹) Mean ± 1σ | 2DVD (m s⁻¹) Mean ± 1σ |
|---|---|---|---|
| 0.6 to 0.8 | 2.6 to 3.5 | $2.6 \pm 0.6$ | $2.5 \pm 0.8$ |
| 0.8 to 1.0 | 3.5 to 4.3 | $3.4 \pm 0.6$ | $3.3 \pm 0.9$ |
| 1.0 to 1.2 | 4.3 to 4.9 | $4.2 \pm 0.6$ | $4.1 \pm 0.9$ |
| 1.2 to 1.4 | 4.9 to 5.5 | $4.9 \pm 0.5$ | $5.0 \pm 0.8$ |
| 1.4 to 1.6 | 5.5 to 6.1 | $5.6 \pm 0.5$ | $5.7 \pm 0.7$ |
| 1.6 to 1.8 | 6.1 to 6.6 | $6.1 \pm 0.4$ | $6.2 \pm 0.7$ |
| 1.8 to 2.0 | 6.6 to 7.0 | $6.7 \pm 0.4$ | $6.6 \pm 0.8$ |
| | | | |
| D range (mm) (Huntsville) | Expected (m/s) at 0 km | MPS (m/s) Mean ± Std_dev | 2DVD (m/s) Mean ± Std. dev |
| 0.6 to 0.8 | 2.5 to 3.3 | $2.6 \pm 0.6$ | $2.5 \pm 0.7$ |
| 0.8 to 1.0 | 3.3 to 4.0 | $3.4 \pm 0.5$ | $3.3 \pm 0.7$ |
| 1.0 to 1.2 | 4.0 to 4.6 | $4.2 \pm 0.6$ | $4.1 \pm 0.8$ |
| 1.2 to 1.4 | 4.6 to 5.2 | $4.9 \pm 0.4$ | $4.9 \pm 0.7$ |
| 1.4 to 1.6 | 5.2 to 5.7 | $5.4 \pm 0.4$ | $5.4 \pm 0.6$ |
| 1.6 to 1.8 | 5.7 to 6.1 | $6.0 \pm 0.3$ | $5.8 \pm 0.6$ |
| 1.8 to 2.0 | 6.1 to 6.5 | $6.5 \pm 0.4$ | $6.3 \pm 0.5$ |

The histograms in Fig. 1(b,e) show good agreement between 2DVD and MPS for 1 mm and 1.5 mm drop sizes, respectively, with respect to the mode, symmetry, spectral width and lack of skewness in the distributions. For the 1 mm size histogram, the mean is 3.8 m s⁻¹ while the spectral width or standard deviation from MPS data is 0.6 m s⁻¹. The corresponding coefficient of variation (ratio of standard deviation to mean) is 15.7%. The finite bin width used (0.9-1.1 mm) causes a corresponding fall speed "spread" of around 0.6 m s⁻¹ which is clearly a significant contributor to the measured coefficient of

variation. Similar comments apply to the fall speed histogram for the 1.5 mm size shown in Fig. 1c. The definition of sub- or super-terminal fall speeds by (*Montero-Martinez et al.,* 2009) is based on fall speeds that are, respectively, less than 0.7 times the mean value or greater than 1.3 times the mean value (i.e., exceeding 30% threshold on either side of the mean terminal fall speed). From examining the 1 mm size fall speed histogram there is negligible evidence of occurrences with fall speeds $< 2.66$ m s$^{-1}$ (sub) or $> 4.94$ m s$^{-1}$ (super). Similar comment also applies for the 1.5 mm size based on the corresponding histogram.

[Figure]

*Figure 1.* *(a) Fall velocity versus diameter (D). The contoured frequency of occurrence from 2DVD data is shown in color (log scale). The mean fall velocity and ±1σ standard deviation bars are from MPS. The dark dashed line is from the fit to the laboratory data of Gunn and Kinzer (1949) and the* *purple* *line is the same except corrected for the altitude of Greeley, CO (1.4 km*

*MSL). (b) Relative frequency histograms of fall velocity for the 0.5±0.1 mm and 1±0.1 mm bins.(c) as in (b) except for the 0.7±0.1 mm and 1.5±0.1 mm bins.*

The histogram from MPS for the 0.5 mm sizes shows positive skewness with mean of 1.8 m s$^{-1}$, spectral width of 0.65 m s$^{-1}$ and corresponding coefficient of variation nearly doubling to 35% (relative to the 1 mm size histogram). The finite bin width (0.4-0.6 mm) causes a corresponding fall speed "spread" of 0.4 m s$^{-1}$ which contributes to the measured coefficient of variation. Nevertheless, it is not possible to rule out the low frequency of occurrence of sub- or super-terminal fall speeds, respectively, less than 1.26 m s$^{-1}$ or exceeding 2.34 m s$^{-1}$ based on our data. Examination of the MPS-based fall speed histogram for the 0.7 mm size indicates negative skewness. As with the 0.5 mm drops it is not possible to rule out the occurrences of fall speeds < 1.8 m s$^{-1}$ or > 3.4 m s$^{-1}$, i.e., sub- or super-terminal fall speeds.

2.2 Fall Speeds from Huntsville, Alabama

The first Huntsville event occurred on 11 April 2016 and consisted of precipitation associated with the mesoscale vortex of a developing squall line that moved across northern Alabama between 1800 and 2300 UTC and produced over 25 mm of rainfall in the Huntsville area. Figure 2(a) shows the ambient 10-m height wind speeds (3 s and 5-min averaged) recorded at the site. Maximum speeds were less than 5 m s$^{-1}$ and wind gusts were light. As no direct *in situ* measurement of turbulence was available we use the approach by (*Garrett and Yuter,* 2014) who estimate the difference between the maximum wind speed, or gust, that was sampled every 3 s, and the average wind speed derived from successive 5 min intervals. The estimated turbulent intensity is proportional to $E = (\text{Gusts} - \text{AverageWind})^2/2$. Figure 2(b) shows the $E$ values which were small (maximum $E < 0.4$ m$^2$ s$^{-2}$) and indicative of low turbulence. Also, shown in Fig. 2(b) is the 2DVD-based time series of rainfall rate (R) averaged over 3 mins; the maximum R is around 10 mm h$^{-1}$.

[Figure]

*Figure 2: (a) 3-s raw and 5-min averaged wind speeds at 10-m height. (b) turbulent intensity estimates E, and 3-min averaged R.*

Figure 3(a) shows the fall velocity versus D comparison between the two instruments while panels (b,c) show the histograms for the 0.5 and 1 mm, and 0.7 and 1.5 mm sizes, respectively. Similar to the Greeley event, the mean fall speed agreement between both instruments in the overlap region is excellent (see Table 1) and consistent with the FT fit to the Gunn-Kinzer laboratory data. As in Fig. 1(a), the MPS data in Fig. 3(a) is in excellent agreement with FT fit for sizes < 0.5 mm.

The 0.5 and 1 mm histogram shapes in Fig. 3(b) are quite similar to the Greeley case shown in Fig. 1(b). The mean and standard deviations from the MPS data for the 0.5 and 1 mm bins are, respectively, [2 ± 0.62] and [3.88 ± 0.44] m s$^{-1}$. The values for the 0.7 and 1.5 mm bins are, respectively, [2.6 ± 0.6] and [5.4 ± 0.4] m s$^{-1}$. There is negligible evidence of sub- or super-terminal fall speed occurrences based on the 1 and 1.5 mm histograms. The comments made earlier with respect to Fig. 1(b,c) of the Greeley event for the 0.5 and 0.7 mm histograms are also applicable here, i.e., we cannot rule out the occasional occurrences of sub- or super-terminal fall speeds based on our data.

[Figure]

*Figure 3. (a) as in Fig. 1(a) except for 11 April 2016 event. The dashed line is fit to Gunn-Kinzer at sea level. (b,c) as in Fig. 1(b,c) except for 11 April 2016 event.*

The second case considered is from 30 November 2016 wherein a supercell passed over the instrumented site from 0300-0330 UTC producing about 15 mins later a long-lived EF-2 tornado. Strong winds were recorded at the site with 5-min averaged speeds reaching 10-12 m s$^{-1}$ between 0320-0330 and E values in the range to 7-8 m$^2$ s$^{-2}$ indicating strong turbulence (Fig. 4a,b). The rain rates peaked at 70 mm h$^{-1}$ during this time (Fig. 4b). About 3 h later several squall-line type storm cells passed over the site from 0700-0900 UTC again with strong winds but considerably lower E values 2-4 m$^2$ s$^{-2}$ and maximum R of 80 mm h$^{-1}$. After 1000 UTC the E values were much smaller (< 0.5 m$^2$ s$^{-2}$) indicating calm conditions. The peak R is also smaller at 30 mm h$^{-1}$ at 1000 UTC.

Figure 4 panels (c), (d) and (e) show the mean and ±1σ of the fall speeds from the 2DVD for the 1.3, 2 and 3 mm drop sizes, respectively. The MPS data are not shown

here since during this event it was located outside the DFIR on its turntable and we did not want to confuse the wind-effects between the two instruments. It is clear from Fig. 4(c) that during the supercell passage (0300-0330 UTC) the mean fall speed for 1.3 mm drops decreases (from 5 to 3.5 m s$^{-1}$) and the standard deviation increases (from 0.5 to 1.5 m s$^{-1}$). The histogram shapes also show increasing negative skewness (not shown). The same trend can be seen for the subsequent squall-line rain cell passage from 0700-0900 UTC. Similar trends are noted in panels (d) and and less so in panel (e).

[Figure]

*Figure 4*. (a) as in Fig. 2(a) except for 30 Nov 2016 event. (b) as in Fig. 2(b). (c) mean and ±1σ standard deviation of fall speeds from 2DVD for 1.3±0.1 mm sizes. (d,e) as in (c) except for 2±0.1 and 3±0.1 mm sizes, respectively.

To expand on this observed correlation, Fig. 5 shows scatterplots of the mean fall speed and standard deviation versus $E$ for the 1.3 mm drops (panels a,b), while panels (c,d) and (e,f) show the same but for the 2 and 3 mm drops, respectively. The mean fall speed decreases with increasing $E$ nearly linearly for E>1 $m^2$ $s^{-2}$ but less so for the 3 mm size drops (*Stout et al.,* 1995). This decrease relative to *Gunn-Kinzer* terminal fall speeds is termed as "sub-terminal" and our data is in general agreement with (*Montero-Martinez and Garcia-Garcia* 2016) who found an increase in the numbers of sub-terminal drops with sizes between 1-2 mm under windy conditions using a 2D-Precipitation probe with resolution of 200 μm (similar to 2DVD) but without a wind fence. The standard deviation of fall speeds ($\sigma_f$) versus $E$ is shown in panels 5 (b,d,f). When $E$>1 $m^2$ $s^{-2}$, the $\sigma_f$ is nearly constant at 1.5 m $s^{-1}$ for both 1.3 and 2 mm drop sizes and constant at 1 m $s^{-1}$ for the 3 mm size. For $E$<1, the $\sigma_f$ is more variable and essentially uncorrelated with $E$. From the discussion related to Fig. 1(b,c) and 3(b,c), $\sigma_f$ values exceeding approximately 0.5 m $s^{-1}$ can be attributed to physical, not instrumental or finite bin width effects (see, also, Table 1). Thus, the fall speed distributions are considerably broadened when $E$>1 $m^2$ $s^{-2}$ due to increasing turbulence levels which is again consistent with the findings of (*Montero-Martinez and Garcia-Garcia*, 2016) as well as those of (*Garett and Yuter*, 2014). The latter observations, however, were of graupel fall speeds in winter precipitation using a multi-angle snowflake camera (*Garrett et al.,* 2012).

[Figure]

*Figure 5.* (a,b) mean fall speed and standard deviation, respectively, versus E for 1.3 mm sizes. (c,d) same but for 2 mm sizes. (e,f) same but for 3 mm.

**3 Discussion and Conclusions**

We have reported on raindrop fall speed distributions using a high resolution (50 μm) droplet spectrometer (MPS) collocated with moderate resolution (170 μm) 2DVD (with both instruments inside a DFIR wind shield) to cover the entire size range (from 0.1 mm onwards) expected in natural rain. Turbulence intensity (*E*) was derived from wind/gust

data at 10-m height following (*Garrett and Yuter*, 2014). For low turbulent intensities ($E < 0.4$ m$^2$ s$^{-2}$), in the overlap region of the two instruments (0.7-2 mm), the mean fall speeds were in excellent agreement with each other for both the Greeley, CO and Huntsville, AL sites giving high confidence in the quality of the measurements. For D<0.5 mm and down to 0.1 mm, the mean fall speeds from MPS from both sites were in remarkable agreement with the (*Foote and du Toit*, 1969) fit to the laboratory data of (*Gunn and Kinzer*, 1949). In the overlap region, the mean fall speeds from the two instruments were in excellent agreement with the FT fit for the Huntsville site (no altitude adjustment required) and good agreement for the Greeley site (after adjustment for altitude of 1.4 km). For D>2 mm, the mean fall speeds from 2DVD were in excellent agreement with the FT fit at both sites.

Our histograms of fall speeds for 1 and 1.5 mm sizes under low turbulence intensity conditions ($E < 0.4$ m$^2$ s$^{-2}$) from both MPS and 2DVD were in good agreement and did not show any evidence of either sub- or super-terminal speeds, rather the histograms were symmetric with mean close to the Gunn-Kinzer terminal velocity with no significant broadening over that ascribed to instrument and/or finite bin width effects. (Note: sub-terminal implies fall speeds < 0.7 times the terminal fall speed whereas super-terminal implies > 1.3 times terminal value; Montero-Martinez et al., 2009). However, for the 0.5 and 0.7 mm sizes, from the histogram of fall speeds using the MPS under the same conditions occasional occurrences of both sub- and super-terminal fall speeds, after accounting for instrumental and finite bin width effects, cannot be ruled out.

The only comparable earlier study is by (*Montero-Martinez et al.*, 2009) who used collocated 2D-cloud and precipitation probes (2D-C, 2D-P) but restricted their data to calm wind conditions. Their main conclusion was that the distribution of the ratio of the measured fall speed to the terminal fall speed for 0.44 mm size, while having a mode at 1 m s$^{-1}$ was strongly positively skewed with tails extending to 5 m s$^{-1}$ especially at high rain rates. In our data for the 0.5 and 0.7 mm sizes shown in Fig. 1(b,c) and 3(b,c), no such strong positive skewness was observed in the fall speed histograms, and the corresponding ratio of MPS-measured fall speeds to terminal values does not exceed 1.5 to 2.

Another study by *Larsen et al.,* (2014) appears to confirm the ubiquitous existence of super-terminal fall speeds for sizes < 1 mm using different instruments one of which was a 2DVD similar to the one used in this study. However, it is well-known that "mis-matched" drops cause erroneous fall speed estimates from 2DVD for drops <0.5 mm (*Schoenhuber et al.,* 2008; Appendix in *Huang et al.,* 2010; *Bernauer et al.,* 2015). It is not clear if (*Larsen et al.,* 2014) accounted for this problem in their analysis. In addition, their 2DVD was not located within a DFIR-like wind shield.

In a later study using only the 2D-P probe, (*Montero-Martinez and Garcia-Garcia*, 2016) found sub-terminal fall speeds and broadened distributions under windy conditions for 1-2 mm sizes in general agreement with our results using the 2DVD. *Stout et al.*, (1995) simulated the motion of drops subject to non-linear drag in isotropic turbulence and determined that there would be a significant reduction of the average drop settling velocity (relative to terminal velocity) of greater that 35% for drops around 2 mm size when the ratio of *rms* velocity fluctuations (due to turbulence) relative to drop terminal velocity is around 0.8. Whereas we did not have a direct measure of the *rms* velocity fluctuations, the proxy for turbulence intensity (*E*) related to wind gusts during supercell passage (very large *E* around 7 $m^2$ $s^{-2}$) and two squall-line passages (moderate *E* between 2-5 $m^2$ $s^{-2}$) clearly showed a significant reduction in mean fall speeds of 25-30% relative to terminal speed for 1.3 and 2 mm sizes (and less so for 3 mm drops), with significant broadening of the fall speed distributions relative to calm conditions by nearly a factor of 1.5 to 2.

While our dataset is limited to three events they cover a wide range of rain rates, wind conditions and two different climatologies. One caveat is that the response of the DFIR wind shield to ambient winds in terms of producing subtle vertical air motions near the sensor area is yet to be evaluated as future work.  Analysis of further events with direct measurement of turbulent intensity, for example using a 3D-sonic anemometer at the height of the sensor, would be needed to generalize our findings.

**Data Availability**

Data used in this paper can be accessed from:

ftp://lab.chill.colostate.edu/pub/kennedy/merhala/Bringi_et_al_2017_GRL_datasets/

**Competing interests**

VNB and MT declare they have no conflict of interest. DB is employed by Droplet Measurements Technologies, Inc. (Longmont, Colorado, USA) who manufacture the Meteorological Particle Spectrometer used in this study.

**Acknowledgements**

Two of the authors (VNB and MT) acknowledge support from the U.S. National Science Foundation via grant AGS-1431127. The assistance of Dr. Patrick Gatlin of NASA/MSFC is gratefully acknowledged. Prof. Kevin Knupp and Mr. Carter Hulsey of the University of Alabama in Huntsville  processed the wind data.